# A Unified Framework for Alternating Offline Model Training and Policy Learning

**Shentao Yang[1],  Shujian Zhang[1],  Yihao Feng[2],  Mingyuan Zhou[1]**
[1]The University of Texas at Austin    [2]Salesforce Research
shentao.yang@mccombs.utexas.edu    szhang19@utexas.edu
yihaof@salesforce.com  mingyuan.zhou@mccombs.utexas.edu

## Abstract

In offline model-based reinforcement learning (offline MBRL), we learn a dynamic model from historically collected data, and subsequently utilize the learned model and fixed datasets for policy learning, without further interacting with the environment. Offline MBRL algorithms can improve the efficiency and stability of policy learning over the model-free algorithms. However, in most of the existing offline MBRL algorithms, the learning objectives for the dynamic models and the policies are isolated from each other. Such an *objective mismatch* may lead to inferior performance of the learned agents. In this paper, we address this issue by developing an iterative offline MBRL framework, where we maximize a lower bound of the true expected return, by alternating between dynamic-model training and policy learning. With the proposed unified model-policy learning framework, we achieve competitive performance on a wide range of continuous-control offline reinforcement learning datasets. Source code is released at https://github.com/Shentao-YANG/AMPL_NeurIPS2022.

## 1   Introduction

Offline reinforcement learning (offline RL) [1, 2], where the agents are trained from static and pre-collected datasets, avoids direct interactions with the underlying real-world environment during the learning process. Unlike traditional online RL, whose success largely depends on simulator-based trial-and-error [*e.g.*, 3, 4], offline RL enables training policies for real-world applications, where it is infeasible or even risky to collect online experimental data, such as robotics, advertisement, or dialog systems [*e.g.*, 5–8]. Though promising, it remains challenging to train agents under the offline setting, due to the discrepancy between the distribution of the offline data and the state-action distribution induced by the current learning policy. With such a discrepancy, directly transferring standard online off-policy RL methods [*e.g.*, 9–11] to the offline setting tends to be problematic [12, 13], especially when the offline data cannot sufficiently cover the state-action space [14]. To tackle this issue in offline RL, recent works [*e.g.*, 15–17] propose to approximate the policy-induced state-action distribution by leveraging a learned dynamic model to draw imaginary rollouts. These additional synthetic rollouts help mitigate the distributional discrepancy and stabilize the policy-learning algorithms under the offline setting.

Most of the prior offline model-based RL (MBRL) methods [*e.g.*, 16, 18–21], however, first *pretrain* a one-step forward dynamic model via maximum likelihood estimation (MLE) on the offline dataset, and then use the learned model to train the policy, without *further improving the dynamic model* during the policy learning process. As a result, the objective function used for model training (*e.g.*, MLE) and the objective of model utilization are *unrelated* with each other. Specifically, the model is trained to be "simply a mimic of the world," but is used to improve the performance of the learned policy [22–24]. Though such a training paradigm is historically rooted [25, 26], this issue of *objective*

36th Conference on Neural Information Processing Systems (NeurIPS 2022).

*mismatch* in the model training and model utilization has been identified as problematic in recent works [23, 24]. In offline MBRL, this issue is exacerbated, since the learned model can hardly be globally accurate, due to the limited amount of offline data and the complexity of the control tasks.

Motivated by the objective-mismatch issue, we develop an iterative offline MBRL method, alternating between training the dynamic model and the policy to maximize a lower bound of the true expected return. This lower bound, leading to a weighted MLE objective for the dynamic-model training, is relaxed to a tractable regularized objective for the policy learning. To train the dynamic model by the proposed objective, we need to estimate the marginal importance weights (MIW) between the offline-data distribution and the stationary state-action distribution of the current policy [27, 28]. This estimation tends to be unstable by standard approaches [*e.g.*, 29, 30], which require saddle-point optimization. Instead, we propose a simple yet stable fixed-point-style method for MIW estimation, which can be directly incorporated into our alternating training framework. With these considerations, our method, offline Alternating Model-Policy Learning (AMPL), performs competitively on a wide range of continuous-control offline RL datasets in the D4RL benchmark [31]. These empirical results and ablation study show the efficacy of our proposed algorithmic designs.

## 2 Background

**Markov decision process and offline RL.** A Markov decision process (MDP) is denoted by $\mathcal{M} = (\mathbb{S}, \mathbb{A}, P, r, \gamma, \mu_0)$, where $\mathbb{S}$ is the state space, $\mathbb{A}$ the action space, $P(s' \mid s, a) : \mathbb{S} \times \mathbb{S} \times \mathbb{A} \to [0, 1]$ the environmental dynamic, $r(s, a) : \mathbb{S} \times \mathbb{A} \to [-r_{max}, r_{max}]$ the reward function, $\gamma \in [0, 1)$ the discount factor, and $\mu_0(s) : \mathbb{S} \to [0, 1]$ the initial state-distribution.

For any policy $\pi(a \mid s)$, we denote its state-action distribution at timestep $t \geq 0$ as $d_{\pi,t}^P(s, a) \triangleq \Pr(s_t = s, a_t = a \mid s_0 \sim \mu_0, a_t \sim \pi, s_{t+1} \sim P, \forall t \geq 0)$. The (discounted) stationary state-action distribution of $\pi$ is denoted as $d_{\pi,\gamma}^P(s, a) \triangleq (1 - \gamma) \sum_{t=0}^{\infty} \gamma^t d_{\pi,t}^P(s, a)$.

Denote $Q_{\pi}^P(s, a) = \mathbb{E}_{\pi,P} \left[ \sum_{t=0}^{\infty} \gamma^t r(s_t, a_t) \mid s_0 = s, a_0 = a \right]$ as the action-value function of policy $\pi$ under the dynamic $P$. The goal of RL is to find a policy $\pi$ maximizing the expected return

$$J(\pi, P) \triangleq (1 - \gamma) \mathbb{E}_{s \sim \mu_0, a \sim \pi(\cdot \mid s)} \left[ Q_{\pi}^P(s, a) \right] = \mathbb{E}_{(s,a) \sim d_{\pi,\gamma}^P} \left[ r(s, a) \right]. \tag{1}$$

In offline RL, the policy $\pi$ and critic $Q_{\pi}^P$ are typically approximated by parametric functions $\pi_{\phi}$ and $Q_{\theta}$, respectively, with parameters $\phi$ and $\theta$. The critic $Q_{\theta}$ is trained by the Bellman backup

$$\arg \min_{\theta} \mathbb{E}_{(s,a,r,s') \sim \mathcal{D}_{\text{env}}} \left[ \left( Q_{\theta}(s, a) - \left( r(s, a) + \gamma \mathbb{E}_{a' \sim \pi_{\phi}(\cdot \mid s')} \left[ Q_{\theta'}(s', a') \right] \right) \right)^2 \right], \tag{2}$$

where $Q_{\theta'}$ is the target network [12, 13]. The actor $\pi_{\phi}$ is trained in the policy improvement step by

$$\arg \max_{\phi} \mathbb{E}_{s \sim \mathcal{D}_{\text{env}}, a \sim \pi_{\phi}(\cdot \mid s)} \left[ Q_{\theta}(s, a) \right], \tag{3}$$

where $\mathcal{D}_{\text{env}}$ denotes the offline dataset drawn from $d_{\pi_b,\gamma}^P$ [2, 32], with $\pi_b$ being the behavior policy.

**Offline model-based RL.** In offline model-based RL algorithms, the true environmental dynamic $P^*$ is typically approximated by a parametric function $\widehat{P}(s' \mid s, a)$ in some function class $\mathcal{P}$. With the offline dataset $\mathcal{D}_{\text{env}}$, $\widehat{P}$ is trained via the MLE [15, 16, 18] as

$$\arg \max_{\widehat{P} \in \mathcal{P}} \mathbb{E}_{(s,a,s') \sim \mathcal{D}_{\text{env}}} \left[ \log \widehat{P}(s' \mid s, a) \right]. \tag{4}$$

Similarly, the reward function can be approximated by a parametric model $\widehat{r}$ if assumed unknown. With $\widehat{P}$ and $\widehat{r}$, the true MDP $\mathcal{M}$ can be approximated by $\widehat{\mathcal{M}} = (\mathbb{S}, \mathbb{A}, \widehat{P}, \widehat{r}, \gamma, \mu_0)$. We further define $d_{\pi,\gamma}^{P^*}(s, a)$ as the stationary state-action distribution induced by $\pi$ on $P^*$ (or MDP $\mathcal{M}$), and $d_{\pi,\gamma}^{\widehat{P}}(s, a)$ as that on the learned dynamic $\widehat{P}$ (or MDP $\widehat{\mathcal{M}}$). We approximate $d_{\pi_{\phi},\gamma}^{P^*}$ by simulating $\pi_{\phi}$ on $\widehat{\mathcal{M}}$ for a short horizon $h$ starting from state $s \in \mathcal{D}_{\text{env}}$, as in prior work [*e.g.*, 16, 18, 19, 21]. The resulting transitions are stored in a replay buffer $\mathcal{D}_{\text{model}}$, constructed similar to the off-policy RL [33, 9]. To better approximate $d_{\pi_{\phi},\gamma}^{P^*}$, sampling from $\mathcal{D}_{\text{env}}$ in Eqs. (2) and (3) is commonly replaced by sampling from the augmented dataset $\mathcal{D} = f\mathcal{D}_{\text{env}} + (1 - f)\mathcal{D}_{\text{model}}, f \in [0, 1]$, denoting sampling from $\mathcal{D}_{\text{env}}$ and $\mathcal{D}_{\text{model}}$ with probabilities $f$ and $1 - f$, respectively. We follow Yu et al. [18] to use $f = 0.5$.

# 3 Offline alternating model-policy learning

Our goal is to derive the objectives for both dynamic-model training and policy learning from a principled perspective. A natural idea is to build a tractable lower bound for $J(\pi, P^*)$, the expected return of the policy $\pi$ under the true dynamic $P^*$, and then alternate between training the policy $\pi$ and the dynamic model $\widehat{P}$ to maximize this lower bound. Indeed, we can construct a lower bound as

$$J(\pi, P^*) \geq J(\pi, \widehat{P}) - |J(\pi, P^*) - J(\pi, \widehat{P})|, \tag{5}$$

where $J(\pi, \widehat{P})$ is the expected return of policy $\pi$ under the learned model $\widehat{P}$. From the right hand side (RHS) of Eq. (5), if the policy evaluation error $|J(\pi, P^*) - J(\pi, \widehat{P})|$ is small, $J(\pi, \widehat{P})$ will be a good proxy for the true expected return $J(\pi, P^*)$. We can empirically estimate $J(\pi, \widehat{P})$ via $\widehat{P}$ and $\pi$.

Further, if a tractable upper bound for $|J(\pi, P^*) - J(\pi, \widehat{P})|$ can be constructed, it can serve as a unified training objective for both dynamic model $\widehat{P}$ and policy $\pi$. We can then alternate between optimizing the dynamic model $\widehat{P}$ and the policy $\pi$ to maximize the lower bound of $J(\pi, P^*)$, *i.e.*, simultaneously minimizing the upper bound of the evaluation error $|J(\pi, P^*) - J(\pi, \widehat{P})|$. This gives us an *iterative, maximization-maximization* algorithm for model and policy learning.

The following theorem indicates a tractable upper bound for $|J(\pi, P^*) - J(\pi, \widehat{P})|$, which can be subsequently relaxed for model training and policy learning.

**Theorem 1.** *Let $P^*$ be the true dynamic and $\widehat{P}$ be the approximate dynamic model. Suppose the reward function $|r(s, a)| \leq r_{\max}$, then we have*

$$\left| J(\pi, P^*) - J(\pi, \widehat{P}) \right| \leq \frac{\gamma \cdot r_{\max}}{\sqrt{2}(1 - \gamma)} \cdot \sqrt{D_\pi(P^*, \widehat{P})},$$

*with* $\quad D_\pi(P^*, \widehat{P}) \triangleq \mathbb{E}_{(s,a) \sim d^{P^*}_{\pi_b, \gamma}} \left[ \omega(s, a) \mathrm{KL} \left( P^*(s' \,|\, s, a) \pi_b(a' \,|\, s') \,||\, \widehat{P}(s' \,|\, s, a) \pi(a' \,|\, s') \right) \right],$

*where $\pi_b$ is the behavior policy, $d^{P^*}_{\pi_b, \gamma}$ is the offline-data distribution, and $\omega(s, a) \triangleq \frac{d^{P^*}_{\pi, \gamma}(s, a)}{d^{P^*}_{\pi_b, \gamma}(s, a)}$ is the marginal importance weight (MIW) between the offline-data distribution and the stationary state-action distribution of the policy $\pi$ [27, 29].*

Detailed proof of Theorem 1 can be found in Appendix B.2.

The KL term in $D_\pi(P^*, \widehat{P})$ indicates the following two principles for model and policy learning:

⋆  For the dynamic model, $s' \sim \widehat{P}(\cdot \,|\, s, a)$ should be close to the true next state $\tilde{s}' \sim P^*(\cdot \,|\, s, a)$, with $(s, a)$ pairs drawn from the stationary state-action distribution of the policy $\pi$. Since we cannot directly draw samples from $d^{P^*}_{\pi, \gamma}$, we reweight the offline data with $\omega(s, a)$. This leads to a weighted KL minimization objective for the model training.

⋆  For the policy $\pi$, the KL term indicates a regularization term, that the tuple $(s', a')$ from the joint conditional distribution $\widehat{P}(s' \,|\, s, a) \pi(a' \,|\, s')$ should be close to the tuple $(\tilde{s}', \tilde{a}')$ from $P^*(\tilde{s}' \,|\, s, a) \pi_b(\tilde{a}' \,|\, \tilde{s}')$. $(\tilde{s}', \tilde{a}')$ is simply a sample from the offline dataset.

Based on the above observations, we can fixed $\pi$ and train the dynamic model $\widehat{P}$ by minimizing $D_\pi(P^*, \widehat{P})$ w.r.t. $\widehat{P}$. Similarly, we can fix the dynamic model $\widehat{P}$ and learn a policy $\pi$ to maximize the lower bound of $J(\pi, P^*)$. This alternating training scheme provides a unified approach for model and policy learning. In the following sections, we discuss how to optimize the dynamic model $\widehat{P}$, the policy $\pi$ , and the MIW $\omega$ under our alternating training farmework.

## 3.1 Dynamic model training

Expanding the KL term in $D_\pi(P^*, \widehat{P})$, we have

$$D_\pi(P^*, \widehat{P}) = \overbrace{\mathbb{E}_{(s,a,s',a') \sim d^{P^*}_{\pi_b, \gamma}} \left[ \omega(s, a) \left( \log P^*(s' \,|\, s, a) + \log \pi_b(a' \,|\, s') - \log \pi(a' \,|\, s') \right) \right]}^{\triangleq ①}$$

$$- \mathbb{E}_{(s,a,s') \sim d^{P^*}_{\pi_b, \gamma}} \left[ \omega(s, a) \log \widehat{P}(s' \,|\, s, a) \right],$$

where the tuple $(s, a, s', a')$ is simply two consecutive state-action pairs in the offline dataset. Further, if the policy $\pi$ is fixed, the term ①is a constant *w.r.t.* $\widehat{P}$. Thus, given the MIW $\omega$, we can optimize $\widehat{P}$ by minimizing the following loss

$$\ell(\widehat{P}) \triangleq -\mathbb{E}_{(s,a,s')\sim d_{\pi_b,\gamma}^{P^*}} \left[ \omega(s,a) \log \widehat{P}(s' \mid s, a) \right], \tag{6}$$

which is an MLE objective weighted by $\omega(s, a)$. We discuss how to estimate $\omega(s, a)$ in Section 3.3.

## 3.2 Policy learning

The lower bound for $J(\pi, P^*)$ implied by Theorem 1 is

$$J(\pi, \widehat{P}) - \tfrac{\gamma \cdot r_{\max}}{\sqrt{2}(1-\gamma)} \cdot \sqrt{D_\pi(P^*, \widehat{P})}, \tag{7}$$

where $J(\pi, \widehat{P})$ can be estimated via the action-value function similar to standard offline MBRL algorithms [*e.g.*, 16, 18, 19]. Thus, when the dynamic model $\widehat{P}$ is fixed, the main difficulty is to estimate the regularizer $D_\pi(P^*, \widehat{P})$ for the policy $\pi$.

When the policy $\pi$ is Gaussian, direct estimation of $D_\pi(P^*, \widehat{P})$ is possible. Empirically, however, it is helpful to learn the policy $\pi$ in the class of implicit distribution, which is a richer distribution class and can better maximize the action-value function. Specifically, given a noise distribution $p_z(z)$, action $a = \pi_\phi(s, z)$ with $z \sim p_z(\cdot)$, where $\pi_\phi$ is a deterministic network.

Unfortunately, we can not directly estimate the KL term in $D_\pi(P^*, \widehat{P})$ if $\pi$ is an implicit policy, since we can only draw samples from $\pi$ but the density is unknown. A potential solution is to use the dual representation of KL divergence $\mathrm{KL}(p \,\|\, q) = \sup_T \mathbb{E}_p[T] - \log(\mathbb{E}_q[e^T])$ [34], which can be estimated with samples from the distributions $p$ and $q$. However, the exponential function therein makes the estimation unstable in practice [30]. We instead use the dual representation of the Jensen–Shannon divergence (JSD) to approximate $D_\pi(P^*, \widehat{P})$, which can be approximately minimized using the GAN structure [35, 36]. Our framework can thus utilize the many stabilization techniques developed in the GAN community (Appendix E.2.1).

Besides, we remove the MIW $\omega(s, a)$ during the policy training since we do not observe its empirical benefits, which will be discussed in Section 4.2. Further applying the replacement of KL with JSD and ignoring the $\sqrt{\cdot}$ for numerical stability, we get an *approximated new regularization* for policy $\pi$:

$$\widetilde{D}_\pi(P^*, \widehat{P}) \triangleq \mathrm{JSD}\left( P^*(s' \mid s, a)\pi_b(a' \mid s')d_{\pi_b,\gamma}^{P^*}(s, a) \,\|\, \widehat{P}(s' \mid s, a)\pi(a' \mid s')d_{\pi_b,\gamma}^{P^*}(s)\pi(a \mid s) \right). \tag{8}$$

Informally speaking, Eq. (8) regularizes the imaginary rollouts of $\pi$ on $\widehat{P}$ towards state-action pairs from the offline dataset. Intuitively, $\widetilde{D}_\pi(P^*, \widehat{P})$ is a more effective regularizer for policy training than the original $D_\pi(P^*, \widehat{P})$, since $\widetilde{D}_\pi(P^*, \widehat{P})$ regularizes action choices at both $s$ and $s'$. Appendix B.3 discusses how we move from $D_\pi(P^*, \widehat{P})$ to $\widetilde{D}_\pi(P^*, \widehat{P})$ in detail.

## 3.3 Marginal importance weight training

A number of methods have been recently proposed to estimate the marginal importance weight $\omega$ [27, 29, 30]. These methods typically require solving a complex saddle-point optimization, casting doubts on their training stability especially when combined with policy learning on continuous-control offline MBRL problems. In this section, we mimic the Bellman backup to derive a fixed-point-style method for estimating the MIW.

Denote the true MIW as $\omega^*(s, a) \triangleq \frac{d_{\pi,\gamma}^{P^*}(s,a)}{d_{\pi_b,\gamma}^{P^*}(s,a)}$, we have $d_{\pi_b,\gamma}^{P^*}(s, a) \cdot \omega^*(s, a) = d_{\pi,\gamma}^{P^*}(s, a)$. Expanding the RHS, $\forall s', a'$,

$$d_{\pi_b,\gamma}^{P^*}(s', a')\omega^*(s', a') = \gamma \sum_{s,a} \pi(a' \mid s')P^*(s' \mid s, a)\omega^*(s, a)d_{\pi_b,\gamma}^{P^*}(s, a) + (1-\gamma)\mu_0(s')\pi(a' \mid s'). \tag{9}$$

The derivation is deferred to Appendix B.4. Therefore, a "*Bellman equation*" for $\omega(s', a')$ is

$$\omega(s', a') = \mathcal{T}\omega(s', a'),$$

$$\mathcal{T}\omega(s', a') \triangleq \frac{\gamma \sum_{s,a} \pi(a' \mid s')P^*(s' \mid s, a)\omega(s, a)d_{\pi_b,\gamma}^{P^*}(s, a) + (1-\gamma)\mu_0(s')\pi(a' \mid s')}{d_{\pi_b,\gamma}^{P^*}(s', a')}.$$

Here $\mathcal{T}$ can be viewed as the "*Bellman operator*" for $\omega$. The update iterate defined by $\mathcal{T}$ has the following convergence property, which is proved in Appendix B.5.

**Proposition 2.** *On finite state-action space, if the current policy $\pi$ is close to the behavior policy $\pi_b$, then the iterate for $\omega$ defined by $\mathcal{T}$ converges geometrically.*

The assumption that $\pi$ is close to $\pi_b$ coincides with the regularization term in the policy-learning objective discussed in Section 3.2.

Unfortunately, the RHS of Eq. (9) is not estimable since we do not know the density values therein. We therefore multiply both sides of Eq. (9) by some test function and subsequently sum over $(s', a')$ on both sides to get a tractable objective that only requires samples from the offline dataset and the initial state-distribution $\mu_0$. It is desired to choose a test function that can better distinguish the difference between the left-hand side (LHS) and the RHS of Eq. (9). A potential choice is the action-value function of the policy $\pi$, due to some primal-dual relationship between the stationary state-action density-(ratio) ($d_{\pi,\gamma}^{P^*}(s, a)$ or $\omega^*(s, a)$) and the action-value function [37–39]. A detailed discussion on the choice of the test function is provided in Appendix C.

Practically we use $Q_\pi^{\widehat{P}}$ as the test function. Note that multiplying both sides of Eq. (9) by the same $Q_\pi^{\widehat{P}}$ does not undermine the convergence property, under mild conditions on Q. Mimicking the Bellman backup to sum over $(s', a')$ on both sides, with the notation $d_{\pi_b,\gamma}^{P^*}(s, a, s') = d_{\pi_b,\gamma}^{P^*}(s, a)P^*(s' \mid s, a)$,

$$\overbrace{\mathbb{E}_{(s,a)\sim d_{\pi_b,\gamma}^{P^*}}\left[\omega^*(s,a)\cdot Q_\pi^{\widehat{P}}(s,a)\right]}^{\ell_1(\omega^*)} = \overbrace{\gamma\mathbb{E}_{\substack{(s,a,s')\sim d_{\pi_b,\gamma}^{P^*}\\ a'\sim\pi(\cdot\,|\,s')}}\left[\omega^*(s,a)\cdot Q_\pi^{\widehat{P}}(s',a')\right] + (1-\gamma)\mathbb{E}_{\substack{s\sim\mu_0(\cdot)\\ a\sim\pi(\cdot\,|\,s)}}\left[Q_\pi^{\widehat{P}}(s,a)\right]}^{\ell_2(\omega^*)}. \quad (10)$$

Thus for a given $\omega$, we can optimize $\omega$ by minimizing the difference between the RHS and the LHS of Eq. (10). For training stability, we use a target network $\omega'(s, a)$ for the RHS, and the final objective for learning $\omega$ is

$$\left(\ell_1(\omega) - \ell_2\left(\omega'\right)\right)^2, \quad (11)$$

where the target network $\omega'(s, a)$ is soft-updated after each gradient step, motivated by $Q_{\boldsymbol{\theta}_j'}$ and $\pi_{\phi'}$.

Our proposed training method is closely related to VPM [40]. By using the MIW $\omega(s, a)$ itself as the test function, VPM leverages the variational power iteration to train MIW iteratively. Instead, our approach uses the current action-value function as the test function, motivated by the primal-dual relationship between the MIW and the action-value function in off-policy evaluation. We compare the empirical performance of several alternative approaches in Section 4.2 and in Table 3 of Appendix A.

### 3.4 Practical implementation

In this section we briefly discuss some implementation details of our offline Alternating Model-Policy Learning (AMPL) method, whose main steps are in Algorithm 1. Further details are in Appendix E.1.

**Dynamic model training.** We adopt common practice in offline MBRL [*e.g.*, 16, 18] to use an ensemble of Gaussian probabilistic networks $\widehat{P}(\cdot \mid s, a)$ and $\hat{r}(s, a)$ to parameterize the stochastic transition and reward. We initialize the dynamic model by standard MLE training, and periodically update the model by minimizing Eq. (6).

**Critic training.** We use the conservative target in the offline RL literature [*e.g.*, 12, 13]:

$$\widetilde{Q}(s,a) \triangleq r(s,a) + \gamma\mathbb{E}_{a'\sim\pi_{\phi'}(\cdot\,|\,s')}[c\min_{j=1,2} Q_{\boldsymbol{\theta}_j'}(s',a') + (1-c)\max_{j=1,2} Q_{\boldsymbol{\theta}_j'}(s',a')], \quad (12)$$

where we set $c = 0.75$. With mini-batch $\mathcal{B}$ sampled from the augmented dataset $\mathcal{D}$, both critic networks are trained as

$$\forall j = 1, 2, \quad \arg\min_{\boldsymbol{\theta}_j} \frac{1}{|\mathcal{B}|}\sum_{(s,a)\in\mathcal{B}} \text{Huber}(Q_{\boldsymbol{\theta}_j}(s,a), \widetilde{Q}(s,a)), \quad (13)$$

where the Huber loss $\text{Huber}(\cdot)$ is used in lieu of the classical MSE for training stability [41].

**Estimating $\widetilde{D}_\pi(P^*, \widehat{P})$.** In $\widetilde{D}_\pi(P^*, \widehat{P})$, using the notations of GAN, we denote the sample from the left distribution of JSD in Eq. (8) by $\mathcal{B}_{\text{true}}$ (*i.e.*, "true" sample), and the sample from the right distribution of JSD by $\mathcal{B}_{\text{fake}}$ (*i.e.*, "fake" sample).

**Algorithm 1** Main steps of AMPL.

---

**Initialize:** Dynamic model $\widehat{P}$ and $\hat{r}$, policy $\pi_\phi$, critics $Q_{\theta_1}$ and $Q_{\theta_2}$, discriminator $D_\psi$, MIW $\omega$.
Initialize $\widehat{P}$ and $\hat{r}$ via the MLE (Eq. (4)).
**for** iteration $\in \{1, \ldots, \texttt{total\_iterations}\}$ **do**
   **if** iteration $\%$ `model_retrain_period` $== 0$ **then**
      Estimate $\omega(s, a)$ via Eq. (11); train $\widehat{P}$ and $\hat{r}$ by weighted MLE (Eq. (6)) with $\omega(s, a)$.
   **end if**
   Rollout synthetic data with $\pi_\phi$, $\widehat{P}$ and $\hat{r}$, and add the data to $\mathcal{D}_{\text{model}}$.
   Sample mini-batch $\mathcal{B} \sim \mathcal{D} = f\mathcal{D}_{\text{env}} + (1-f)\mathcal{D}_{\text{model}}$.
   Optimize $Q_{\theta_1}$, $Q_{\theta_2}$ via Eqs. (12) – (13).
   Train $D_\psi$ to maximize Eq. (16).
   Optimize $\pi_\phi$ by Eq. (17).
**end for**

---

The "true" sample $\mathcal{B}_{\text{true}}$ consists of samples from $\mathcal{D}_{\text{env}}$. The "fake" sample $\mathcal{B}_{\text{fake}}$ is formed by first sampling $s \sim \mathcal{D}$, followed by $a \sim \pi_\phi(\cdot \,|\, s), s' \sim \widehat{P}(\cdot \,|\, s, a), a' \sim \pi_\phi(\cdot \,|\, s')$. Concretely, the "fake" sample $\mathcal{B}_{\text{fake}}$, generator loss $\mathcal{L}_g(\phi)$, and the discriminator loss $\mathcal{L}_D(\psi)$ can be described as

$$\mathcal{B}_{\text{fake}} \triangleq \begin{bmatrix} (s, & a) \\ (s', & a') \end{bmatrix}, \quad (14) \quad \mathcal{L}_g(\phi) \triangleq \frac{1}{|\mathcal{B}_{\text{fake}}|} \sum_{(s,a) \in \mathcal{B}_{\text{fake}}} \left[ \log\left(1 - D_\psi(s, a)\right) \right], \quad (15)$$

$$\mathcal{L}_D(\psi) \triangleq \frac{1}{|\mathcal{B}_{\text{true}}|} \sum_{(s,a) \sim \mathcal{B}_{\text{true}}} \left[ \log D_\psi(s, a) \right] + \frac{1}{|\mathcal{B}_{\text{fake}}|} \sum_{(s,a) \sim \mathcal{B}_{\text{fake}}} \left[ \log\left(1 - D_\psi(s, a)\right) \right], \quad (16)$$

where $\mathcal{L}_g(\phi)$ is the empirical policy-learning regularizer implied by $\widetilde{D}_\pi(P^*, \widehat{P})$.

**Policy training.** Since the constant multiplier in Eq. (7) is unknown, we treat it as a hyperparameter. Adding the regularization term of policy training, our policy optimization objective is

$$\arg\min_\phi -\lambda \cdot \frac{1}{|\mathcal{B}|} \sum_{s \in \mathcal{B}, a \sim \pi_\phi(\cdot \,|\, s)} \left[ \min_{j=1,2} Q_{\theta_j}(s, a) \right] + \mathcal{L}_g(\phi), \quad (17)$$

where the regularization coefficient $\lambda \triangleq \lambda'/Q_{avg}$, with soft-updated $Q_{avg}$ similar to Fujimoto and Gu [42]. We use $\lambda' = 10$ across all datasets in our experiments (Section 4).

## 4 Experiments

In this section, we first evaluate our AMPL on continuous-control offline-RL datasets in Section 4.1. Then we conduct ablation study on Section 4.2 to understand the efficacy of some proposed designs.

### 4.1 Continuous-control results

Our experiments are conducted on a diverse set of datasets in the D4RL benchmark [31], ranging across the Gym-Mojoco, Maze2D, and Adroit domains therein. We use the latest version of the datasets, *i.e.*, "v2" version for the Gym-Mojoco domain and "v1" version for the Maze2D and Adroit domains. Details of our dataset choice are discussed in Appendix E.2.

We compare our AMPL with two DICE methods — AlgaeDICE [43] and OptiDICE [44] — that directly utilize MIW to improve policy learning. We further consider three state-of-the-art (SOTA) offline model-free RL algorithms: CQL [45], FisherBRC [46], and TD3+BC [42]; and three SOTA offline MBRL methods: MOPO [16], COMBO [18], and WMOPO [47]. Experimental details and hyperparameter settings are discussed in Appendix E.2. We run baseline methods using the official implementation under the recommended hyperparameters, except for AlgaeDICE, for which we use the offline version provided by Fu et al. [31]. Table 1 shows the mean and standard deviation of the results of AMPL and baselines over five random seeds.

As shown in Table 1, our AMPL performs comparably well and is relatively stable across the sixteen tested datasets. In particular, AMPL generally performs better than the baselines on the Maze2D and Adroit datasets. The Maze2D and Adroit datasets are considered as more challenging than the MuJoCo datasets [31], since the Maze2D datasets are collected by non-Markovian policies, and the amount of data in the high-dimensional Adroit datasets is limited.

Table 1: Normalized returns for experiments on the D4RL tasks. Mean and standard deviation across five random seeds are reported. High average score and low average rank are desirable. We bold the best result over all methods and underline the best of the model-based methods if different. Here, "hcheetah" denotes "halfcheetah," "med" denotes "medium," "rep" denotes "replay," and "exp" denotes "expert."

| Task Name | AlgaeDICE | OptiDICE | CQL | FisherBRC | TD3+BC | MOPO | COMBO | WMOPO | AMPL |
|---|---|---|---|---|---|---|---|---|---|
| maze2d-large | -2.2 ± 0.6 | 101.7 ± 50.7 | 1.5 ± 6.4 | 0.9 ± 6.4 | 107.1 ± 45.9 | -0.5 ± 2.6 | 138.5 ± 82.2 | 1.8 ± 8.6 | **180.0** ± 39.3 |
| maze2d-med | 2.5 ± 14.2 | **119.4** ± 52.3 | 6.3 ± 9.1 | 16.1 ± 30.3 | 61.4 ± 45.5 | 12.5 ± 18.3 | 103.9 ± 42.1 | 12.7 ± 38.3 | 107.2 ± 45.0 |
| maze2d-umaze | -15.3 ± 0.8 | **114.0** ± 39.7 | 37.5 ± 7.2 | 3.6 ± 16.4 | 38.6 ± 14.4 | -15.4 ± 1.9 | 112.1 ± 56.8 | -11.5 ± 3.1 | 55.8 ± 3.9 |
| hcheetah-med | -0.9 ± 0.7 | 42.0 ± 3.5 | 48.6 ± 0.2 | 47.8 ± 0.3 | 48.0 ± 0.3 | 69.2 ± 3.4 | **73.0** ± 3.6 | 72.0 ± 4.7 | 51.7 ± 0.4 |
| walker2d-med | 1.0 ± 2.1 | 55.7 ± 15.7 | 81.8 ± 1.7 | 80.7 ± 2.1 | 83.0 ± 1.4 | -0.1 ± 0.0 | 0.5 ± 0.9 | 64.8 ± 30.0 | **83.1** ± 1.8 |
| hopper-med | 0.9 ± 0.2 | 57.5 ± 7.5 | 68.0 ± 6.0 | 94.2 ± 4.5 | 59.1 ± 4.5 | 44.8 ± 41.5 | 22.6 ± 38.1 | **99.7** ± 2.9 | 58.9 ± 7.9 |
| hcheetah-med-rep | -3.4 ± 2.3 | 40.5 ± 3.3 | 38.7 ± 18.6 | 34.9 ± 18.2 | 44.5 ± 0.6 | 62.7 ± 7.5 | 66.0 ± 1.8 | **66.4** ± 4.9 | 44.6 ± 0.7 |
| walker2d-med-rep | 0.5 ± 0.6 | 37.3 ± 21.8 | 80.8 ± 4.0 | **84.6** ± 6.7 | 74.7 ± 8.1 | 53.8 ± 34.6 | 60.1 ± 18.3 | 71.9 ± 15.0 | 81.5 ± 3.0 |
| hopper-med-rep | 1.5 ± 0.7 | 28.1 ± 12.4 | **96.0** ± 4.7 | 94.4 ± 1.9 | 58.9 ± 19.7 | 84.8 ± 30.0 | 53.6 ± 29.5 | 93.8 ± 8.5 | 91.1 ± 9.5 |
| hcheetah-med-exp | -1.8 ± 2.9 | 66.7 ± 25.8 | 53.8 ± 13.6 | **94.5** ± 0.6 | 91.4 ± 5.2 | 64.0 ± 20.8 | 52.5 ± 32.0 | 68.8 ± 37.5 | 90.2 ± 2.0 |
| walker2d-med-exp | -0.2 ± 0.1 | 79.4 ± 16.4 | 110.3 ± 0.6 | 109.3 ± 0.1 | **110.4** ± 0.6 | -0.2 ± 0.0 | 1.1 ± 0.6 | 98.8 ± 16.2 | 107.7 ± 2.1 |
| hopper-med-exp | 2.2 ± 1.9 | 52.6 ± 9.3 | 79.3 ± 20.5 | **110.0** ± 4.0 | 98.7 ± 9.6 | 15.6 ± 7.4 | 63.7 ± 51.9 | 62.6 ± 26.4 | 76.0 ± 15.7 |
| pen-human | -3.0 ± 0.8 | -0.9 ± 3.1 | -1.5 ± 2.8 | -0.1 ± 3.0 | 1.6 ± 2.0 | -1.6 ± 2.6 | 5.9 ± 8.7 | -2.1 ± 1.4 | **20.6** ± 10.7 |
| pen-cloned | -2.6 ± 1.3 | -0.8 ± 3.1 | 39.5 ± 23.6 | -1.2 ± 3.8 | 6.3 ± 4.6 | 5.2 ± 10.2 | 23.2 ± 19.8 | -3.0 ± 2.4 | **57.4** ± 19.2 |
| pen-exp | -1.3 ± 2.4 | 0.4 ± 7.3 | 119.3 ± 15.8 | 2.2 ± 1.1 | 104.2 ± 40.6 | 35.4 ± 20.9 | 57.1 ± 42.6 | 10.4 ± 20.9 | **138.3** ± 6.6 |
| door-exp | 0.1 ± 0.0 | 86.1 ± 23.2 | 94.0 ± 15.8 | 22.9 ± 28.8 | -0.3 ± 0.0 | -0.0 ± 0.1 | -0.2 ± 0.2 | -0.1 ± 0.1 | **96.3** ± 9.8 |
| Average Score | -1.4 | 55 | 59.6 | 49.7 | 61.7 | 26.9 | 52.1 | 44.2 | **83.8** |
| Average Rank | 8.5 | 5.5 | 4.1 | 4.3 | 3.9 | 6.4 | 4.8 | 5.0 | **2.6** |

On the Maze2D datasets, the behavioral-cloning (BC) style algorithms, such as FisherBRC, are likely to fail, since these methods use Markovian policy to approximate the non-Markovian behavior policy. Meanwhile, OptiDICE performs relatively well on the Maze2D datasets, mainly because it uses a Gaussian-mixture policy for BC to alleviate this approximation difficulty. This advanced BC policy still offers little help on the higher-dimensional Adroit datasets, due to the challenge of learning the behavior policy. Similarly, TD3+BC performs well on the relatively lower-dimensional Maze2D datasets, but does not work well on the Adroit domain, since the action-space MSE-regularizer in TD3+BC may be insufficient for the high-dimensional tasks. WMOPO also shows a performance drop in these two challenging task domains. The environmental dynamics on these two domains are complex, and thus may not be accurately learned to support WMOPO's full-trajectory model rollouts, which are used to estimate its MIW. By contrast, our AMPL does not require BC or long model-rollouts, leading to stable performance across task domains. Moreover, AMPL shows generally better performance than MOPO and COMBO, which use fixed pretrained dynamic models without considering the mismatched model objectives.

Compared with the DICE-based methods AlgaeDICE and OptiDICE, our AMPL shows generally better performance. This may indicate that maximizing a lower bound of the true expected return can be a more effective framework than explicit stationary-distribution correction, since maximizing the lower bound is more directly related to RL's goal of maximizing the policy performance.

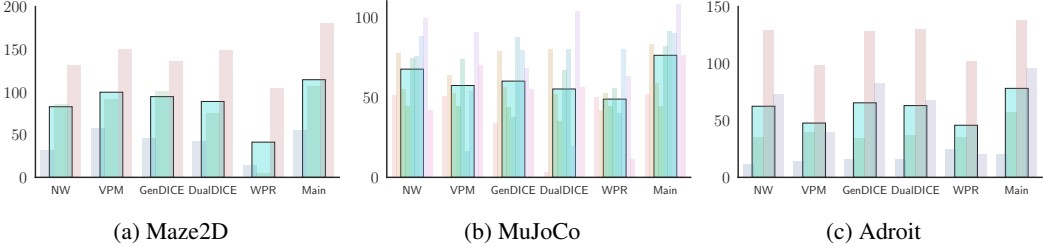

(a) Maze2D      (b) MuJoCo      (c) Adroit

Figure 1: Scores of each method in the ablation study (Section 4.2) on each domain of datasets. The faded bars show the scores on each datasets within the stated domain, averaged over five seeds, where each color corresponds to a dataset. The highlighted bar shows the average score on the entire domain. Label "Main" refers to the results of our main method in Section 4.1. Detailed numbers are on Table 2.

## 4.2 Ablation study

**(a):** *Does the MIW-weighted model (re)training help the performance?*

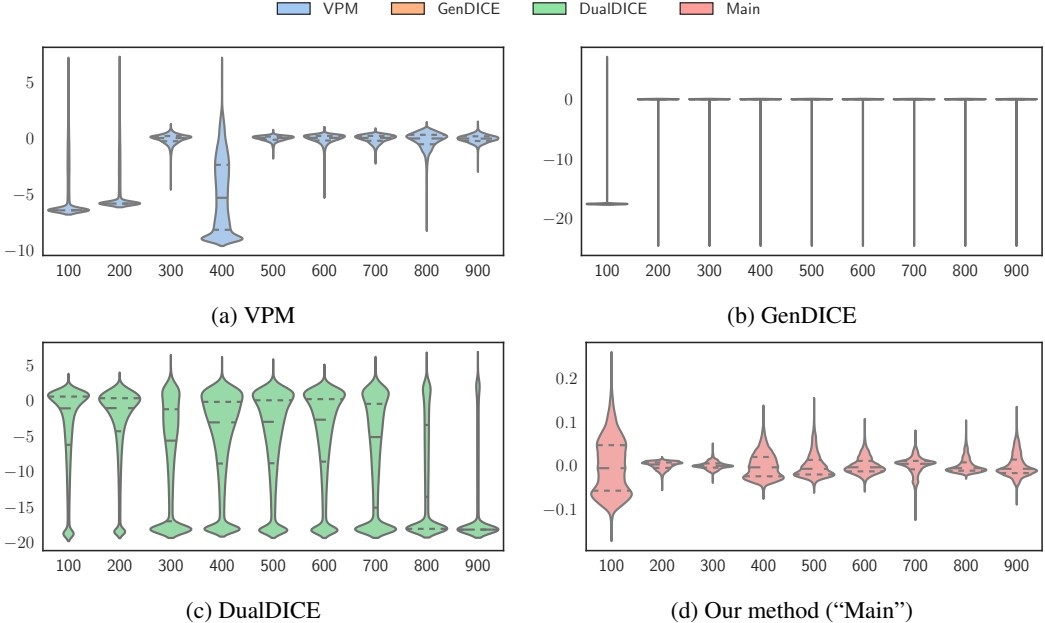

|  | VPM | GenDICE | DualDICE | Main |

(a) VPM

(b) GenDICE

(c) DualDICE

(d) Our method ("Main")

Figure 2: Distribution plots of $\log$ (MIW) of the entire dataset by our method ("Main") and the three alternatives in Section 4.2 (b), on the "walker2d-medium-replay" dataset during the training process. The dash lines represent the quartiles of the distribution. Recall that the MIW $\omega$ is retrained every 100 epochs. The $x$-axis represents the number of epochs $\{100, 200, \ldots, 900\}$, and the $y$-axis the $\log$ (MIW) values. For clarity, we use the normalized MIWs whose mean on the entire dataset is one. We use random seed two for this plot. Other random seeds have similar patterns. We note the different scales in the $y$-axis of these four plots.

To verify the effectiveness of our MIW-weighted model (re)training scheme, we compare our AMPL with its variant of training the model only at the beginning using MLE, *i.e.*, No Weights (dubbed as NW). As shown in Figure 1, the performance of the NW variant is generally worse than the main method (dubbed as Main) on all three domains. The performance difference is especially significant on the Maze2D domain. In this domain, the policy is required to "stitch together collected subtrajectories to find the shortest path to the evaluation goal" [31]. As a result, the state-action distribution induced by a good policy should be different from the distribution of the offline-data. Table 2 in Appendix A confirms the benefit of the MIW-weighted model (re)training. Indeed, on the majority of datasets, the NW variant not only shows worse performance, but also has a larger standard deviation relative to the mean score.

**(b):** *How does the algorithm perform if we change our MIW estimation method to other approaches?*

We compare our AMPL with its three variants where the MIW is instead estimated by the Variational Power Method [40], GENeralized stationary DIstribution Correction Estimation [30], and Dual stationary DIstribution Correction Estimation [29]. Three variants are simply dubbed as VPM, GenDICE, and DualDICE. For numerical stability, the estimated MIW from these three methods is clipped into $(10^{-8}, 500)$. Implementation details are provided in Appendix E.3.

As shown in Figure 1 and Table 2, the variants with these three alternative MIW estimation methods generally perform worse than our approach. A plausible explanation is that these methods can be unstable to provide good MIW estimates for the model training. Figure 2 shows the distribution plots of $\log$ (MIW) of the entire dataset estimated by our method and by the variants using the three alternative MIW estimation methods, on the "walker2d-medium-replay" dataset. We train the MIW every 100 epochs and ignore the MIW initialization from the plots. Notice that the MIWs obtained by these four methods have very different scales, which indicates the instability of the alternative methods compared with ours.

As shown on Figure 2b, the MIWs estimated by the GenDICE variant degenerate, in a sense that most of the MIWs are $\approx 0$ at the begining and are 1 later on, with long tails on the MIW distributions. This may explain the relatively bad performance of GenDICE. Compared with GenDICE, DualDICE provides more diverse MIW estimates, which leads to its relatively better performance. Unfortunately,

Figure 2c shows that, for the DualDICE variant, the distribution of MIWs gradually concentrates on very small and very large values, which indicates the degeneration of the MIWs. The VPM variant performs the best among these three alternatives, mainly because it provides relatively better MIW estimates in the second half of the training process. However, Figure 2a shows that the MIWs from the VPM variant are poorly-distributed at the beginning, though relatively better later on. Our method provides well-behaved MIW estimates in the whole training process, which may explain our better result. As shown by Figure 2d, the MIWs from our method are well-shaped and concentrate around the mean 1 [1] over the entire training process. In general, the well-shaping of the estimated MIW is important, since the Cramer-Rao lower bound of the mean-square-error for OPE is related to the square of the density ratio [38, 48].

Besides, on the Maze2D domain, these three variants using the alternative MIW estimation methods perform generally better than the NW variant discussed in Question (**a**), aligning with the benefit of the MIW-weighted model (re)training scheme. Generally speaking, incorporating the MIW can help model training in offline MBRL.

(**c**): *What is the performance of a weighted regularizer for policy learning implied by Theorem 1?*

We compare our AMPL with its variant where the policy regularizer is weighted by the MIW $\omega(s, a)$, as suggested by Theorem 1. This Weighted Policy Regularizer variant is dubbed as WPR. As shown in both Figure 1 and Table 2, the WPR variant underperforms our main method. This is because when we estimate the regularization term in WPR, we incorporate weights into the minimax optimization of the policy and the discriminator, which may bring additional instability.

Table 3 of Appendix A provides additional ablation study on the performance of several alternatives.

# 5 Related work

**Offline MBRL.** Most of the existing offline MBRL works focus on policy learning under a given model trained by MLE. These works typically constrain the learned policy to avoid visiting regions where the discrepancy between the true and the learned dynamic is large, thus reducing the policy evaluation error of using the learned dynamic [16, 19, 49, 50]. Besides, some recent works [51, 52] also adopt a GAN-style stochastic policy for its flexibility. Rather than using a fixed MLE-trained model, we derive an objective that trains both the policy and the dynamic model toward maximizing a lower bound of true expected return (simultaneously minimizing the policy evaluation error $|J(\pi, P^*) - J(\pi, \widehat{P})|$). Several recent works also propose to enhance model training, such as training a reverse dynamic model to encourage conservative model-based imaginations [53], learning a balanced state-action representation for the model to mitigate the distributional discrepancy [54, 55], and using advanced dynamic-model architecture of the GPT-Transformer [56–59] to achieve accurate predictions [60]. These methods are orthogonal to ours and may further improve the performance.

**Objective mismatch in MBRL.** Our AMPL is related to prior works on online and offline MBRL that mitigate the mismatched model objectives. In online (off-policy) MBRL, Lambert et al. [23] identify the mismatched objectives between the MLE model-training and the model's usage of improving the control performance. This paper, however, only proposes an impractical weighted learning loss that requires the optimal trajectory. Eysenbach et al. [24] design an objective to jointly optimize the policy and the model in online MBRL. Voelcker et al. [61] propose a per-sample diagonal scaling matrix for training a deterministic model in online RL, which improves on Farahmand et al. [62] and Farahmand [63]. In offline MBRL, D'Oro et al. [64] propose a weighting scheme to learn the transition model. This method requires the behavior policy and estimates the trajectory-wise importance-sampling weight, which is known to suffer from the "curse of horizon" [27]. Thus, this method may not scale to complex high-dimensional offline RL tasks. Voloshin et al. [65] propose a minimax objective for model learning, where the model is optimized over function classes of the MIW and the state-value function. Most similar to our work, Hishinuma and Senda [47] also use a MIW-weighted MLE objective for model training. They use a different version of the MIW estimated via multiple *full-trajectory* rollouts in the learned model. This strategy may suffer from the inaccuracy of the learned model [66–68] and can be time-consuming. In this paper, we develop a fixed-point style method to train the MIW that only requires samples from the offline dataset and the initial state-distribution, which can be more stable and efficient.

---

[1] 0 in the log (MIW) plots.

**Off-policy evaluation (OPE).** OPE [29, 30, 48, 69–74] is a well-studied problem with several lines of research. A natural way is trajectory-wise importance sampling (IS) [75]. This, however, suffers from its variance, which can grow exponentially with the horizon length [27, 76]. Marginal importance sampling methods [27] are developed to address this problem by estimating the IS ratio between two stationary distributions, leading to estimators whose variance are polynomial *w.r.t.* horizon [77]. Though prior works can also be used to estimate the MIW, they may not be directly applicable to our framework, since they either require knowing the behavior policy [27, 37], or need to assume the MIW lying in the space of RKHS [28]. Another approach to estimate the MIW is through saddle-point optimization [29, 30, 39, 78], which can suffer from its complex optimization problem. In this paper, we derive an fixed-point-style and behavior-agnostic MIW estimation method, which is simple to train and does not require additional assumptions.

## 6   Conclusion

In this paper, we are motivated by the mismatched model objectives in offline MBRL to design an iterative algorithm that alternates between model training and policy optimization. Both the model and the policy are trained to maximize a lower bound of the true expected return. The proposed new algorithm performs competitively with several SOTA baselines.

**Limitation.** Our current approach requires training two additional networks (discriminator $D_\psi$ and MIW $\omega$), and the model is (re)trained periodically to mitigate the objective mismatch, which brings additional computational cost in order to obtain better empirical results.

**Future work.** We plan to conduct a theoretical analysis of our proposed framework, and investigate more efficient ways to unify model training and policy learning.

## Acknowledgments

S. Yang, S. Zhang, and M. Zhou acknowledge the support of NSF IIS 1812699 and 2212418, and the Texas Advanced Computing Center (TACC) for providing HPC resources that have contributed to the research results reported within this paper.

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
