# Appendix

## A  Additional table

Table 2 presents the numerical results for the ablation study in Section 4.2.

Table 2: Normalized returns for experiments on the D4RL tasks. Mean and standard deviation across five random seeds are reported. NW denotes the No-Weights variant in Section 4.2 **(a)**. VPM, GenDICE, and DualDICE denote the variants for the MIW estimation in Section 4.2 **(b)**. WPR denotes the Weighted Policy Regularizer variant in Section 4.2 **(c)**. The results of our main method in Section 4.1 is reported in column Main. Here, "hcheetah" denotes "halfcheetah", "med" denotes "medium", "rep" denotes "replay", and "exp" denotes "expert".

| Task Name | NW | VPM | GenDICE | DualDICE | WPR | Main |
|---|---|---|---|---|---|---|
| maze2d-umaze | $31.3 \pm 25.3$ | $57.8 \pm 29.1$ | $46.2 \pm 23.1$ | $42.6 \pm 19.4$ | $14.4 \pm 21.4$ | $55.8 \pm 3.9$ |
| maze2d-med | $85.4 \pm 28.6$ | $91.1 \pm 54.3$ | $101.3 \pm 50.3$ | $74.8 \pm 42.6$ | $5.0 \pm 8.3$ | $107.2 \pm 45.0$ |
| maze2d-large | $131.4 \pm 63.0$ | $150.2 \pm 64.3$ | $136.1 \pm 63.0$ | $149.2 \pm 32.7$ | $104.3 \pm 48.1$ | $180.0 \pm 39.3$ |
| Average Maze2D | 82.7 | 99.7 | 94.5 | 88.9 | 41.2 | 114.3 |
| hcheetah-med | $51.5 \pm 0.3$ | $50.8 \pm 0.7$ | $33.5 \pm 18.8$ | $3.5 \pm 1.8$ | $50.1 \pm 0.4$ | $51.7 \pm 0.4$ |
| walker2d-med | $77.3 \pm 6.0$ | $63.7 \pm 28.3$ | $78.7 \pm 5.3$ | $79.7 \pm 4.7$ | $41.8 \pm 18.9$ | $83.1 \pm 1.8$ |
| hopper-med | $55.0 \pm 8.9$ | $52.3 \pm 6.6$ | $56.5 \pm 11.3$ | $51.6 \pm 9.1$ | $52.5 \pm 21.6$ | $58.9 \pm 7.9$ |
| hcheetah-med-rep | $44.6 \pm 0.3$ | $44.3 \pm 0.4$ | $43.8 \pm 1.2$ | $34.9 \pm 18.1$ | $44.4 \pm 0.4$ | $44.6 \pm 0.7$ |
| walker2d-med-rep | $74.6 \pm 4.2$ | $73.5 \pm 7.3$ | $37.8 \pm 32.6$ | $66.9 \pm 29.6$ | $55.8 \pm 14.3$ | $81.5 \pm 3.0$ |
| hopper-med-rep | $75.4 \pm 19.1$ | $16.6 \pm 20.0$ | $87.7 \pm 16.6$ | $79.7 \pm 19.8$ | $40.1 \pm 11.1$ | $91.1 \pm 9.5$ |
| hcheetah-med-exp | $88.0 \pm 5.1$ | $54.0 \pm 43.2$ | $79.0 \pm 18.6$ | $19.4 \pm 35.9$ | $80.1 \pm 5.9$ | $90.2 \pm 2.0$ |
| walker2d-med-exp | $99.1 \pm 7.9$ | $90.3 \pm 30.2$ | $68.3 \pm 32.3$ | $103.9 \pm 4.3$ | $62.9 \pm 17.6$ | $107.7 \pm 2.1$ |
| hopper-med-exp | $41.7 \pm 16.9$ | $69.8 \pm 9.6$ | $54.7 \pm 22.5$ | $56.2 \pm 26.5$ | $11.3 \pm 7.6$ | $76.0 \pm 15.7$ |
| Average MuJoCo | 67.5 | 57.3 | 60.0 | 55.1 | 48.8 | 76.1 |
| pen-human | $11.4 \pm 8.6$ | $13.8 \pm 12.1$ | $15.7 \pm 9.7$ | $16.4 \pm 11.9$ | $24.8 \pm 14.7$ | $20.6 \pm 10.7$ |
| pen-cloned | $35.3 \pm 15.7$ | $39.5 \pm 15.8$ | $34.4 \pm 17.2$ | $37.3 \pm 16.4$ | $35.5 \pm 13.8$ | $57.4 \pm 19.2$ |
| pen-exp | $129.6 \pm 15.4$ | $98.1 \pm 33.7$ | $128.6 \pm 16.2$ | $130.4 \pm 16.0$ | $102.0 \pm 36.3$ | $138.3 \pm 6.6$ |
| door-exp | $73.5 \pm 30.9$ | $39.3 \pm 45.2$ | $83.1 \pm 25.9$ | $68.0 \pm 35.0$ | $20.7 \pm 16.4$ | $96.3 \pm 9.8$ |
| Average Adroit | 62.4 | 47.7 | 65.4 | 63.0 | 45.8 | 78.2 |
| Average All | 69.1 | 62.8 | 67.8 | 63.4 | 46.6 | 83.8 |

Table 3 provides additional ablation study on several building blocks of our main method. In Table 3, we test the following variants to demonstrate the effectiveness of our framework.

- **Main** denotes the results of our main method in Section 4.1.
- **NW** denotes the No-Weights variant in Section 4.2 **(a)**.
- **WPR** denotes the Weighted Policy Regularizer variant in Section 4.2 **(c)**.
- **KL-Dual** denotes the variant changing the JSD regularizer in Main to the $\mathrm{KL}(P^*(s' \,|\, s,a)\pi_b(a' \,|\, s') \,||\, \widehat{P}(s' \,|\, s,a)\pi(a' \,|\, s'))$ term in Theorem 1, where $(s,a) \sim d^{P^*}_{\pi_b,\gamma}$ and the KL term is estimated via the dual representation $\mathrm{KL}(p \,||\, q) = \sup_T \left\{ \mathbb{E}_p[T] - \log(\mathbb{E}_q[e^T]) \right\}$ [34].
- **KL-Dual+WPR** denotes the variant changing the JSD regularizer in Main to the weighted KL term $D_\pi(P^*, \widehat{P})$ in Theorem 1.
- **Gaussian** denotes the variant changing the implicit policy in Main to the Gaussian policy.
- **No Reg.** denotes the variant removing the proposed regularizer in the policy learning.
- **No model-rollout** denotes the variant of no rollout data in policy learning.
- **Rew. Test** denotes the variant of using the estimated reward function as the test function when training the MIW $\omega$.

Table 3: Normalized returns for experiments on the D4RL tasks. Mean and standard deviation across five random seeds are reported.

**Main** denotes the results of our main method in Section 4.1. **NW** denotes the No-Weights variant in Section 4.2 **(a)**. **WPR** denotes the Weighted Policy Regularizer variant in Section 4.2 **(c)**. **KL-Dual** denotes the variant changing the JSD regularizer in Main to the KL term in Theorem 1. **KL-Dual+WPR** denotes the variant changing the JSD regularizer in Main to the weighted KL term $D_\pi(P^*, \widehat{P})$ in Theorem 1. **Gaussian** denotes the variant changing the implicit policy in Main to the Gaussian policy. **No Reg.** denotes the variant removing the proposed regularizer in the policy learning. **No model-rollout** denotes the variant of no rollout data in policy learning. **Rew. Test** denotes the variant of using the estimated reward function as the test function when training the MIW $\omega$. Here, "hcheetah" denotes "halfcheetah," "med" denotes "medium," "rep" denotes "replay," and "exp" denotes "expert."

| Task Name | Main | NW | WPR | KL-Dual | KL-Dual+WPR | Gaussian | No Reg. | No model-rollout | Rew. Test |
|---|---|---|---|---|---|---|---|---|---|
| maze2d-umaze | $55.8 \pm 3.9$ | $31.3 \pm 25.3$ | $14.4 \pm 21.4$ | $-3.3 \pm 18.6$ | $-0.9 \pm 16.5$ | $41.5 \pm 26.3$ | $-13.9 \pm 1.3$ | $48.4 \pm 14.7$ | $34.9 \pm 24.3$ |
| maze2d-medium | $107.2 \pm 45.0$ | $85.4 \pm 28.6$ | $5.0 \pm 8.3$ | $16.9 \pm 17.9$ | $4.8 \pm 23.2$ | $59.5 \pm 29.5$ | $83.3 \pm 30.1$ | $120.7 \pm 41.9$ | $85.6 \pm 45$ |
| maze2d-large | $180.0 \pm 39.3$ | $131.4 \pm 63.0$ | $104.3 \pm 48.1$ | $-1.0 \pm 4.0$ | $-0.5 \pm 7.4$ | $106.1 \pm 51.0$ | $0.6 \pm 0.7$ | $121.0 \pm 26.5$ | $143.8 \pm 44$ |
| Average Maze2D | 114.3 | 82.7 | 41.2 | 4.2 | 1.1 | 69.0 | 23.3 | 96.7 | 88.1 |
| hcheetah-med | $51.7 \pm 0.4$ | $51.5 \pm 0.3$ | $50.1 \pm 0.4$ | $16.2 \pm 13.0$ | $28.3 \pm 2.9$ | $49.2 \pm 0.4$ | $30.5 \pm 26.3$ | $51.9 \pm 0.6$ | $51.5 \pm 0.6$ |
| walker2d-med | $83.1 \pm 1.8$ | $77.3 \pm 6.0$ | $41.8 \pm 18.9$ | $0.3 \pm 1.5$ | $2.8 \pm 3.4$ | $72.5 \pm 6.7$ | $3.5 \pm 4.4$ | $83.3 \pm 4.4$ | $77.8 \pm 8.4$ |
| hopper-med | $58.9 \pm 7.9$ | $55.0 \pm 8.9$ | $52.5 \pm 21.6$ | $7.6 \pm 3.4$ | $6.9 \pm 7.0$ | $52.1 \pm 3.7$ | $6.2 \pm 9.0$ | $42.8 \pm 22.8$ | $57.3 \pm 9.7$ |
| hcheetah-med-rep | $44.6 \pm 0.7$ | $44.6 \pm 0.3$ | $44.4 \pm 0.4$ | $23.1 \pm 5.9$ | $18.9 \pm 8.0$ | $41.1 \pm 0.7$ | $67.3 \pm 2.7$ | $41.9 \pm 1.7$ | $44.6 \pm 0.2$ |
| walker2d-med-rep | $81.5 \pm 3.0$ | $74.6 \pm 4.2$ | $55.8 \pm 14.3$ | $2.3 \pm 4.4$ | $1.1 \pm 2.5$ | $54.1 \pm 5.8$ | $10.6 \pm 8.9$ | $13.8 \pm 13.4$ | $72.8 \pm 10.4$ |
| hopper-med-rep | $91.1 \pm 9.5$ | $75.4 \pm 19.1$ | $40.1 \pm 11.1$ | $6.0 \pm 5.7$ | $5.5 \pm 7.3$ | $71.3 \pm 19.7$ | $9.9 \pm 6.5$ | $85.8 \pm 16.8$ | $79.3 \pm 20.6$ |
| hcheetah-med-exp | $90.2 \pm 2.0$ | $88.0 \pm 5.1$ | $80.1 \pm 5.9$ | $20.7 \pm 11.1$ | $19.1 \pm 9.4$ | $85.0 \pm 2.9$ | $7.5 \pm 12.6$ | $79.0 \pm 7.2$ | $88.5 \pm 4.5$ |
| walker2d-med-exp | $107.7 \pm 2.1$ | $99.1 \pm 7.9$ | $62.9 \pm 17.6$ | $0.3 \pm 1.0$ | $1.3 \pm 2.7$ | $90.6 \pm 6.6$ | $2.6 \pm 0.8$ | $104.7 \pm 3.4$ | $103 \pm 6.9$ |
| hopper-med-exp | $76.0 \pm 15.7$ | $41.7 \pm 16.9$ | $11.3 \pm 7.6$ | $3.1 \pm 1.4$ | $4.5 \pm 3.3$ | $59.1 \pm 25.5$ | $12.4 \pm 9.6$ | $48.7 \pm 17.4$ | $59.8 \pm 26$ |
| Average MuJoCo | 76.1 | 67.5 | 48.8 | 8.8 | 9.8 | 63.9 | 16.7 | 61.3 | 70.5 |
| pen-human | $20.6 \pm 10.7$ | $11.4 \pm 8.6$ | $24.8 \pm 14.7$ | $9.3 \pm 4.0$ | $4.7 \pm 7.6$ | $8.5 \pm 5.6$ | $1.6 \pm 5.4$ | $8.4 \pm 9.0$ | $12.2 \pm 10.6$ |
| pen-cloned | $57.4 \pm 19.2$ | $35.3 \pm 15.7$ | $35.5 \pm 13.8$ | $22.4 \pm 12.5$ | $35.7 \pm 12.8$ | $39.3 \pm 16.4$ | $39.1 \pm 21.3$ | $3.5 \pm 6.2$ | $37.5 \pm 16.6$ |
| pen-exp | $138.3 \pm 6.6$ | $129.6 \pm 15.4$ | $102.0 \pm 36.3$ | $31.2 \pm 15.5$ | $30.1 \pm 18.3$ | $102.0 \pm 14.9$ | $5.2 \pm 7.6$ | $109.2 \pm 40.9$ | $133 \pm 12.7$ |
| door-exp | $96.3 \pm 9.8$ | $73.5 \pm 30.9$ | $20.7 \pm 16.4$ | $0.1 \pm 1.4$ | $0.2 \pm 0.4$ | $20.3 \pm 21.2$ | $-0.3 \pm 0.1$ | $53.9 \pm 41.4$ | $86 \pm 24.3$ |
| Average Adroit | 78.2 | 62.4 | 45.8 | 15.8 | 17.7 | 42.5 | 11.4 | 43.8 | 67.2 |
| Average All | 83.8 | 69.1 | 46.6 | 9.7 | 10.2 | 59.5 | 16.6 | 63.6 | 73.0 |

Apart from the discussion in Section 4.2 **(a)** and Section 4.2 **(c)** on the **NW** and the **WPR** variants. We see that changing the proposed JSD regularizer in Section 3.2 to the KL-dual-based regularizers breaks the policy learning. This may be related to the unstable estimation of KL-dual discussed in Section 3.2. Changing the implicit policy to the Gaussian policy generally leads to worse performance. The performance difference is especially significant on the Maze2D and Adroit datasets. This aligns with the observation in Yang et al. [51] that a uni-model Gaussian policy may be insufficient to capture the necessary multiple action modes in the Maze2D and Adroit datasets, while an implicit policy is likely to be capable. Removing the proposed regularization term breaks the policy learning, showing the efficacy of the proposed regularizer. Removing rollout data in the policy learning generally leads to worse performance and larger standard deviations. This shows the benefit of adding the model-rollout data, that they can mitigate the off-policy issue by taking into account the rollouts of the learned policy. Finally, compared with the main method that uses the action-value function as the test function, the **Rew. Test** variant generally has worse performance and larger standard deviations. As discussed in Appendix C, using the (estimated) reward function as the test function loses the nice mathematical properties, and the resulting objective for training the MIW $\omega$ may not be easy to optimize.

# B Proofs

## B.1 Preliminary on discounted stationary state-action distribution

Recall that the discounted visitation frequency for a policy $\pi$ on MDP $\mathcal{M}$ with transition $P^*$ is defined as

$$d_{\pi,\gamma}^{P^*}(s,a) \triangleq (1-\gamma) \sum_{t=0}^{\infty} \gamma^t \Pr\left(s_t = s, a_t = a \mid \mu_0, \pi, P^*\right).$$

Denote $\boldsymbol{T}_\pi^*(s' \mid s) = p_\pi\left(s_{t+1} = s' \mid s_t = s\right) = \sum_{a \in \mathbb{A}} P^*\left(s_{t+1} = s' \mid s_t = s, a_t = a\right) \pi\left(a \mid s\right).$

From Liu et al. [27], Lemma 3, for $d_{\pi,\gamma}^{P^*}(s)$ we have,

$$\gamma \sum_s \boldsymbol{T}_\pi^*(s' \mid s) d_{\pi,\gamma}^{P^*}(s) - d_{\pi,\gamma}^{P^*}(s') + (1-\gamma)\mu_0(s') = 0, \quad \forall s' \in \mathbb{S},$$

where $\mu_0$ is the initial-state distribution. Multiply $\pi(a' \mid s')$ on both sides, we get, $\forall (s', a') \in \mathbb{S} \times \mathbb{A}$,

$$\gamma \sum_s \boldsymbol{T}_\pi^*(s' \mid s) d_{\pi,\gamma}^{P^*}(s)\pi(a' \mid s') - d_{\pi,\gamma}^{P^*}(s', a') + (1-\gamma)\mu_0(s')\pi(a' \mid s') = 0$$

$$\Longleftrightarrow \gamma \sum_{s,a} P^*(s' \mid s, a)\, d_{\pi,\gamma}^{P^*}(s, a)\pi(a' \mid s') - d_{\pi,\gamma}^{P^*}(s', a') + (1-\gamma)\mu_0(s')\pi(a' \mid s') = 0$$

We can also multiply $\widetilde{\pi}(a' \mid s')$ on both sides to get, $\forall (s', a') \in \mathbb{S} \times \mathbb{A}$,

$$\gamma \sum_s \boldsymbol{T}_\pi^*(s' \mid s) d_{\pi,\gamma}^{P^*}(s)\widetilde{\pi}(a' \mid s') - d_{\pi,\gamma}^{P^*}(s')\widetilde{\pi}(a' \mid s') + (1-\gamma)\mu_0(s')\widetilde{\pi}(a' \mid s') = 0.$$

Denote $d_{\pi,\gamma}^{P^*}(s, a, s') \triangleq d_{\pi,\gamma}^{P^*}(s)\pi(a \mid s)P^*(s' \mid s, a)$. For any integrable function $f(s,a)$, we multiply both sides of the above equation by $f(s', a')$ and summing over $s', a'$, we get

$$\gamma \mathbb{E}_{\substack{(s,a,s') \sim d_{\pi,\gamma}^{P^*} \\ a' \sim \widetilde{\pi}(\cdot \mid s')}} [f(s', a')] - \mathbb{E}_{\substack{s \sim d_{\pi,\gamma}^{P^*} \\ a \sim \widetilde{\pi}(\cdot \mid s)}} [f(s, a)] + (1-\gamma)\mathbb{E}_{\substack{s \sim \mu_0 \\ a \sim \widetilde{\pi}(\cdot \mid s)}} [f(s, a)] = 0.$$

For any given bounded function $g(s, a)$, define function $f$ to satisfy

$$f(s, a) = g(s, a) + \gamma \mathbb{E}_{s' \sim P^*(\cdot \mid s, a), a' \sim \pi(\cdot \mid s')} [f(s', a')], \forall s, a,$$

then we have

$$\mathbb{E}_{s \sim \mu_0(\cdot), a \sim \pi(\cdot \mid s)} [f(s, a)] = \mathbb{E}_{s \sim \mu_0(\cdot), a \sim \pi(\cdot \mid s)} \left[ g(s, a) + \gamma \mathbb{E}_{s' \sim P^*(\cdot \mid s, a), a' \sim \pi(\cdot \mid s')} [f(s', a')] \right]$$

$$= \mathbb{E} \left[ \sum_{t=0}^{\infty} \gamma^t g(s_t, a_t) \mid s_0 \sim \mu_0(\cdot), a_t \sim \pi(\cdot \mid s_t), s_{t+1} \sim P^*(\cdot \mid s_t, a_t) \right]$$

$$= (1-\gamma)^{-1} \mathbb{E}_{(s,a) \sim d_{\pi,\gamma}^{P^*}} [g(s, a)],$$

based on the definition of $d_{\pi,\gamma}^{P^*}(s, a)$ stated above.

Indeed, $f(s, a)$ is the action-value function of policy $\pi$ under the reward $g(s, a)$ on MDP $\mathcal{M}$, and can be approximated using neural network under the classical regularity assumptions on $g(s, a)$ and $\mathcal{M}$. Also, under these classical regularity conditions, reward function $g$ and its action-value function $f$ have one-to-one correspondence, with $f$ being the unique solution to the Bellman equation.

### B.2 Proofs of Theorem 1

The following Lemma will be used in the proof of Theorem 1.

**Lemma 3.** *For any* $\widehat{P}(s' \mid s, a), \pi(a' \mid s'), P^*(s' \mid s, a), \pi_b(a' \mid s')$, *we have*

$$\mathrm{KL}\left( P^*(s' \mid s, a)\pi_b(a' \mid s') \mid\mid \widehat{P}(s' \mid s, a)\pi(a' \mid s') \right) \geq \mathrm{KL}\left( P^*(s' \mid s, a) \mid\mid \widehat{P}(s' \mid s, a) \right).$$

*Proof.* Since $\mathrm{KL}(\cdot \mid\mid \cdot) \geq 0$, we have

$$\mathrm{KL}\left( P^*(s' \mid s, a)\pi_b(a' \mid s') \mid\mid \widehat{P}(s' \mid s, a)\pi(a' \mid s') \right)$$

$$= \int P^*(s' \mid s, a)\pi_b(a' \mid s') \log \frac{P^*(s' \mid s, a)\pi_b(a' \mid s')}{\widehat{P}(s' \mid s, a)\pi(a' \mid s')} \, \mathrm{d}(s', a')$$

$$= \int P^*(s' \mid s, a)\pi_b(a' \mid s') \left( \log \frac{P^*(s' \mid s, a)}{\widehat{P}(s' \mid s, a)} + \log \frac{\pi_b(a' \mid s')}{\pi(a' \mid s')} \right) \mathrm{d}(s', a')$$

$$= \int P^*(s' \mid s, a)\pi_b(a' \mid s') \log \frac{\pi_b(a' \mid s')}{\pi(a' \mid s')} \, \mathrm{d}(s', a') + \int P^*(s' \mid s, a) \log \frac{P^*(s' \mid s, a)}{\widehat{P}(s' \mid s, a)} \, \mathrm{d}s'$$

$$= \mathbb{E}_{s' \sim P^*(\cdot \mid s, a)} \left[ \mathrm{KL}\left( \pi_b(a' \mid s') \mid\mid \pi(a' \mid s') \right) \right] + \mathrm{KL}\left( P^*(s' \mid s, a) \mid\mid \widehat{P}(s' \mid s, a) \right)$$

$$\geq \mathrm{KL}\left( P^*(s' \mid s, a) \mid\mid \widehat{P}(s' \mid s, a) \right),$$

as desired. $\qquad\square$

*Remark* 4. It also holds that

$$\mathbb{E}_{(s,a)\sim d_{\pi_b,\gamma}^{P^*}}\left[\text{KL}\left(P^*(s'\,|\,s,a)\pi_b(a'\,|\,s')\,||\,\widehat{P}(s'\,|\,s,a)\pi(a'\,|\,s')\right)\right]$$

$$\leq \text{KL}\left(P^*(s'\,|\,s,a)\pi_b(a'\,|\,s')d_{\pi_b,\gamma}^{P^*}(s,a)\,||\,\widehat{P}(s'\,|\,s,a)\pi(a'\,|\,s')d_{\pi_b,\gamma}^{P^*}(s)\pi(a\,|\,s)\right),$$

by using similar proof steps.

*Proof of Theorem 1.*

$$\left|J(\pi,\widehat{P}) - J(\pi,P^*)\right|$$

$$= \left|(1-\gamma)\mathbb{E}_{\substack{s\sim\mu_0\\a\sim\pi(\cdot|s)}}\left[Q_\pi^{\widehat{P}}(s,a)\right] - \mathbb{E}_{d_{\pi,\gamma}^{P^*}}\left[Q_\pi^{\widehat{P}}(s,a)\right] + \mathbb{E}_{d_{\pi,\gamma}^{P^*}}\left[Q_\pi^{\widehat{P}}(s,a)\right] - \mathbb{E}_{(s,a)\sim d_{\pi,\gamma}^{P^*}}[r(s,a)]\right|$$

$$= \left|(1-\gamma)\mathbb{E}_{\substack{s\sim\mu_0\\a\sim\pi(\cdot|s)}}\left[Q_\pi^{\widehat{P}}(s,a)\right] - \mathbb{E}_{d_{\pi,\gamma}^{P^*}}\left[Q_\pi^{\widehat{P}}(s,a)\right] + \mathbb{E}_{(s,a)\sim d_{\pi,\gamma}^{P^*}}\left[Q_\pi^{\widehat{P}}(s,a) - r(s,a)\right]\right|$$

$$= \left|(1-\gamma)\mathbb{E}_{\substack{s\sim\mu_0\\a\sim\pi(\cdot|s)}}\left[Q_\pi^{\widehat{P}}(s,a)\right] - \mathbb{E}_{d_{\pi,\gamma}^{P^*}}\left[Q_\pi^{\widehat{P}}(s,a)\right] + \gamma\mathbb{E}_{(s,a)\sim d_{\pi,\gamma}^{P^*}}\left[\mathbb{E}_{\substack{s'\sim\widehat{P}(\cdot|s,a)\\a'\sim\pi(\cdot|s')}}\left[Q_\pi^{\widehat{P}}(s',a')\right]\right]\right|$$

$$= \left|-\gamma\mathbb{E}_{(s,a)\sim d_{\pi,\gamma}^{P^*}}\left[\mathbb{E}_{\substack{s'\sim P^*(\cdot|s,a)\\a'\sim\pi(\cdot|s')}}\left[Q_\pi^{\widehat{P}}(s',a')\right]\right] + \gamma\mathbb{E}_{(s,a)\sim d_{\pi,\gamma}^{P^*}}\left[\mathbb{E}_{\substack{s'\sim\widehat{P}(\cdot|s,a)\\a'\sim\pi(\cdot|s')}}\left[Q_\pi^{\widehat{P}}(s',a')\right]\right]\right|$$

$$= \gamma\left|\mathbb{E}_{(s,a)\sim d_{\pi,\gamma}^{P^*}}\left[\mathbb{E}_{\substack{s'\sim P^*(\cdot|s,a)\\a'\sim\pi(\cdot|s')}}\left[Q_\pi^{\widehat{P}}(s',a')\right] - \mathbb{E}_{\substack{s'\sim\widehat{P}(\cdot|s,a)\\a'\sim\pi(\cdot|s')}}\left[Q_\pi^{\widehat{P}}(s',a')\right]\right]\right|$$

$$= \gamma\left|\mathbb{E}_{(s,a)\sim d_{\pi,\gamma}^{P^*}}\left[\mathbb{E}_{s'\sim P^*(\cdot|s,a)}\left[V_\pi^{\widehat{P}}(s')\right] - \mathbb{E}_{s'\sim\widehat{P}(\cdot|s,a)}\left[V_\pi^{\widehat{P}}(s')\right]\right]\right| \tag{18}$$

$$\leq \gamma\mathbb{E}_{(s,a)\sim d_{\pi,\gamma}^{P^*}}\left\{\left|\mathbb{E}_{s'\sim P^*(\cdot|s,a)}\left[V_\pi^{\widehat{P}}(s')\right] - \mathbb{E}_{s'\sim\widehat{P}(\cdot|s,a)}\left[V_\pi^{\widehat{P}}(s')\right]\right|\right\}$$

$$\leq \frac{\gamma\cdot r_{\max}}{1-\gamma}\mathbb{E}_{(s,a)\sim d_{\pi,\gamma}^{P^*}}\left\{\sup_{V\in\mathcal{V}}\left|\mathbb{E}_{s'\sim P^*(\cdot|s,a)}[V(s')] - \mathbb{E}_{s'\sim\widehat{P}(\cdot|s,a)}[V(s')]\right|\right\}$$

$$= \frac{\gamma\cdot r_{\max}}{1-\gamma}\mathbb{E}_{(s,a)\sim d_{\pi,\gamma}^{P^*}}\left[\text{TV}\left(P^*(\cdot|s,a)||\widehat{P}(\cdot|s,a)\right)\right]$$

$$\leq \frac{\gamma\cdot r_{\max}}{1-\gamma}\mathbb{E}_{(s,a)\sim d_{\pi,\gamma}^{P^*}}\left[\sqrt{\frac{1}{2}\text{KL}\left(P^*(\cdot|s,a)||\widehat{P}(\cdot|s,a)\right)}\right]$$

$$\leq \frac{\gamma\cdot r_{\max}}{\sqrt{2}\cdot(1-\gamma)}\sqrt{\mathbb{E}_{(s,a)\sim d_{\pi,\gamma}^{P^*}}\left[\text{KL}\left(P^*(\cdot|s,a)||\widehat{P}(\cdot|s,a)\right)\right]}$$

$$\leq \frac{\gamma\cdot r_{\max}}{\sqrt{2}(1-\gamma)}\sqrt{\mathbb{E}_{(s,a)\sim d_{\pi,\gamma}^{P^*}}\left[\text{KL}\left(P^*(s'\,|\,s,a)\pi_b(a'\,|\,s')\,||\,\widehat{P}(s'\,|\,s,a)\pi(a'\,|\,s')\right)\right]}$$

$$= \frac{\gamma\cdot r_{\max}}{\sqrt{2}(1-\gamma)}\sqrt{\mathbb{E}_{(s,a)\sim d_{\pi_b,\gamma}^{P^*}}\left[\omega(s,a)\text{KL}\left(P^*(s'\,|\,s,a)\pi_b(a'\,|\,s')\,||\,\widehat{P}(s'\,|\,s,a)\pi(a'\,|\,s')\right)\right]}$$

where $\mathcal{V}$ is the set of functions bounded by 1, $\omega(s,a) \triangleq \frac{d_{\pi,\gamma}^{P^*}(s,a)}{d_{\pi_b,\gamma}^{P^*}(s,a)}$ is the marginal importance weight (MIW). Here we use the assumption that $\forall s,a,|r(s,a)|\leq r_{\max}$, and hence $|V_\pi^{\widehat{P}}(\cdot)|\leq\frac{r_{\max}}{1-\gamma}$. We use Lemma 3 to introduce $\pi$ and $\pi_b$ into $\text{KL}\left(\cdot\,||\,\cdot\right)$. $\square$

### B.3 Derivation of Eq. (8)

*Derivarion of Eq. (8).* We follow the literature [*e.g.*, 79, 80] to assume that $\forall s, a, \omega(s, a) \le \omega_{\max}$ for some unknown finite constant $\omega_{\max}$. Using Remark 4 and this assumption, we have

$$
\begin{aligned}
D_\pi(P^*, \widehat{P}) &= \mathbb{E}_{(s,a) \sim d^{P^*}_{\pi_b, \gamma}} \left[ \omega(s, a) \mathrm{KL} \left( P^*(s' \,|\, s, a) \pi_b(a' \,|\, s') \,\|\, \widehat{P}(s' \,|\, s, a) \pi(a' \,|\, s') \right) \right] \\
&\le \omega_{\max} \cdot \mathbb{E}_{(s,a) \sim d^{P^*}_{\pi_b, \gamma}} \left[ \mathrm{KL} \left( P^*(s' \,|\, s, a) \pi_b(a' \,|\, s') \,\|\, \widehat{P}(s' \,|\, s, a) \pi(a' \,|\, s') \right) \right] \\
&\le \omega_{\max} \cdot \mathrm{KL} \left( P^*(s' \,|\, s, a) \pi_b(a' \,|\, s') d^{P^*}_{\pi_b, \gamma}(s, a) \,\|\, \widehat{P}(s' \,|\, s, a) \pi(a' \,|\, s') d^{P^*}_{\pi_b, \gamma}(s) \pi(a \,|\, s) \right) \\
&\approx \omega_{\max} \cdot \mathrm{JSD} \left( P^*(s' \,|\, s, a) \pi_b(a' \,|\, s') d^{P^*}_{\pi_b, \gamma}(s, a) \,\|\, \widehat{P}(s' \,|\, s, a) \pi(a' \,|\, s') d^{P^*}_{\pi_b, \gamma}(s) \pi(a \,|\, s) \right) \\
&\triangleq \omega_{\max} \cdot \widetilde{D}_\pi(P^*, \widehat{P}),
\end{aligned}
$$

where we approximate the KL divergence by the JSD. $\qquad\square$

### B.4 Derivation of Eq. (9)

*Derivarion of Eq. (9).* We have, $\forall (s, a) \in \mathbb{S} \times \mathbb{A}$,

$$
\begin{aligned}
& d^{P^*}_{\pi_b, \gamma}(s, a) \cdot \omega(s, a) = d^{P^*}_{\pi, \gamma}(s, a) \iff \\
& d^{P^*}_{\pi_b, \gamma}(s', a') \cdot \omega(s', a') = d^{P^*}_{\pi, \gamma}(s', a') \\
&= \gamma \sum_s T^*_\pi(s' \,|\, s) d^{P^*}_{\pi, \gamma}(s) \pi(a' \,|\, s') + (1 - \gamma) \mu_0(s') \pi(a' \,|\, s') \\
&= \gamma \sum_{s,a} P^*(s' \,|\, s, a) d^{P^*}_{\pi, \gamma}(s, a) \pi(a' \,|\, s') + (1 - \gamma) \mu_0(s') \pi(a' \,|\, s') \\
&= \gamma \sum_{s,a} P^*(s' \,|\, s, a) \omega(s, a) d^{P^*}_{\pi_b, \gamma}(s, a) \pi(a' \,|\, s') + (1 - \gamma) \mu_0(s') \pi(a' \,|\, s'),
\end{aligned}
\tag{19}
$$

where $T^*_\pi(s' \,|\, s) = \sum_{a \in \mathbb{A}} P^*(s' \,|\, s, a) \pi(a \,|\, s)$ is the state-transition kernel. Here we change $(s, a)$ into $(s', a')$ in the "$\iff$" for notation simplicity. $\qquad\square$

### B.5 Proof of Proposition 2

*Proof of Proposition 2.* With the assumed conditions, we would like to show that $\mathcal{T}$ is a contraction mapping under $\|\cdot\|_\infty$, *i.e.*, $\|\mathcal{T}\omega - \mathcal{T}u\|_\infty \le c \cdot \|\omega - u\|_\infty$, for all MIWs $\omega, u : \mathbb{S} \times \mathbb{A} \to \mathbb{R}$, for

some constant $c < 1$. We have,

$$\|\mathcal{T}\omega - \mathcal{T}u\|_\infty$$

$$= \max_{s',a'} |(\mathcal{T}\omega - \mathcal{T}u)(s', a')| = \max_{s',a'} |\mathcal{T}\omega(s', a') - \mathcal{T}u(s', a')|$$

$$= \max_{s',a'} \left| \frac{\gamma \sum_{s,a} \pi(a'\,|\,s') P^*(s'\,|\,s,a) d_{\pi_b,\gamma}^{P^*}(s,a)(\omega(s,a) - u(s,a))}{d_{\pi_b,\gamma}^{P^*}(s', a')} \right|$$

$$\leq \max_{s',a'} \frac{\gamma \sum_{s,a} \pi(a'\,|\,s') P^*(s'\,|\,s,a) d_{\pi_b,\gamma}^{P^*}(s,a) |\omega(s,a) - u(s,a)|}{d_{\pi_b,\gamma}^{P^*}(s', a')}$$

$$\leq \max_{s',a'} \frac{\gamma \sum_{s,a} \pi(a'\,|\,s') P^*(s'\,|\,s,a) d_{\pi_b,\gamma}^{P^*}(s,a)}{d_{\pi_b,\gamma}^{P^*}(s', a')} \cdot \|\omega - u\|_\infty$$

$$= \left\{ \max_{s',a'} \frac{\gamma \sum_{s,a} \pi(a'\,|\,s') P^*(s'\,|\,s,a) d_{\pi_b,\gamma}^{P^*}(s,a)}{\gamma \sum_{s,a} \pi_b(a'\,|\,s') P^*(s'\,|\,s,a) d_{\pi_b,\gamma}^{P^*}(s,a) + (1-\gamma)\mu_0(s')\pi_b(a'\,|\,s')} \right\} \cdot \|\omega - u\|_\infty \quad (20)$$

$$= \left\{ \max_{s',a'} \frac{\pi(a'\,|\,s')}{\pi_b(a'\,|\,s')} \cdot \frac{\gamma \sum_{s,a} P^*(s'\,|\,s,a) d_{\pi_b,\gamma}^{P^*}(s,a)}{\gamma \sum_{s,a} P^*(s'\,|\,s,a) d_{\pi_b,\gamma}^{P^*}(s,a) + (1-\gamma)\mu_0(s')} \right\} \cdot \|\omega - u\|_\infty$$

$$= \left\{ \max_{s',a'} \frac{\pi(a'\,|\,s')}{\pi_b(a'\,|\,s')} \cdot \underbrace{\left( 1 - \frac{(1-\gamma)\mu_0(s')}{\gamma \sum_{s,a} P^*(s'\,|\,s,a) d_{\pi_b,\gamma}^{P^*}(s,a) + (1-\gamma)\mu_0(s')} \right)}_{\triangleq\, c(s') < 1} \right\} \cdot \|\omega - u\|_\infty$$

$$= \left\{ \max_{s',a'} \frac{\pi(a'\,|\,s')}{\pi_b(a'\,|\,s')} \cdot c(s') \right\} \cdot \|\omega - u\|_\infty \triangleq c \cdot \|\omega - u\|_\infty.$$

If the current policy $\pi$ is close to the behavior policy $\pi_b$, in a sense that $\frac{\pi(a'\,|\,s')}{\pi_b(a'\,|\,s')}$ is close to 1 on the entire finite state-action space, specifically,

$$\forall\, s', a', \quad \frac{\pi(a'\,|\,s')}{\pi_b(a'\,|\,s')} < \frac{1}{c(s')}.$$

Then $c = \max_{s',a'} \left\{ \frac{\pi(a'\,|\,s')}{\pi_b(a'\,|\,s')} \cdot c(s') \right\} < 1$. Plug into Eq. (20), we have $\|\mathcal{T}\omega - \mathcal{T}u\|_\infty \leq c \cdot \|\omega - u\|_\infty$ for $c < 1$. Thus $\mathcal{T}$ is a $c$-contraction mapping under $\|\cdot\|_\infty$. It follows that there exists a unique fixed point $\omega^*$ under the mapping $\mathcal{T}$, such that $\omega^* = \mathcal{T}\omega^*$. Finally, we have

$$\|\omega_{k+1} - \omega^*\|_\infty = \|\mathcal{T}\omega_k - \mathcal{T}\omega^*\|_\infty \leq c \cdot \|\omega_k - \omega^*\|_\infty \leq \cdots \leq c^{k+1} \cdot \|\omega_0 - \omega^*\|_\infty \to 0, \text{ as } k \to \infty,$$

which shows that the iterate defined by $\mathcal{T}$ converges geometrically. $\qquad\square$

## C Discussion on the choice of test function for MIW

We first discuss the original minimax optimization loss [27, 30, 81] for learning the MIW $\omega(s, a)$. When given an offline dataset $\mathcal{D}_{\text{env}}$, the objective is

$$\min_{\omega \in W} \max_{f \in \mathcal{F}} \left\{ (1-\gamma)\mathbb{E}_{\mu_0, \pi}[f(s, a)] + \mathbb{E}_{(s,a,s') \sim d_{\pi_b,\gamma}^{P^*}, a' \sim \pi(\cdot\,|\,s')} \left[ \gamma\omega(s,a)f(s',a') - \omega(s,a)f(s,a) \right] \right\}, \quad (21)$$

where $f \in \mathcal{F}$ is a test function similar to the discriminator in adversarial training.

A number of prior works on learning $\omega(s, a)$ (*e.g.* DualDICE [29], GenDICE [30]) can be viewed as variants of the above minimax objective, with some additional regularization terms on $\omega(s, a)$. For example, there is an additional $\frac{1}{2}\omega^2$ term in DualDICE to penalize $\omega$ so that the resulting MIW values $\omega(s, a)$ will not be too large. This term thus serves as a regularization.

However, the optimization process for $\omega(s, a)$ in Eq. (21) can be unstable due to the following two reasons: **(1)** the minimax optimization is itself a challenging problem, especially for neural-network-based function approximator; **(2)** when we fix $f(s, a)$ and optimize $\omega(s, a)$, the gradients for $\omega$ that

come from $\gamma\omega(s,a)f(s',a')$ and $\omega(s,a)f(s,a)$ are close to each other, especially when $\gamma$ is close to 1 (*e.g.*, 0.99). This makes the overall gradient for $\omega$ small and unstable, especially when we use stochastic gradient descent to optimize $\omega$. The empirical distributions of the MIW from DualDICE and GenDICE in Fig. 2 illustrate the optimization difficulty of these two methods. Therefore, we wish to get rid of the minimax optimization and stabilize the training process.

Another closely related work that also tries to address the aforementioned issues is VPM [40], where the authors use the MIW $\omega$ itself as the test function $f$, based on the observation that the optimum test function under the $f$-divergence is the MIW; and use the variational power iteration to solve for $\omega$. We refer to Wen et al. [40] for the detailed derivations. Empirically, we can observe in Fig. 2a that the distribution of $\omega(s,a)$ obtained by VPM tends to be more stable than those obtained by DualDICE and GenDICE.

Our choice of the test function is motivated by the following constrained optimization formulation of off-policy evaluation [37, 81]:

$$\min_{Q} \quad (1-\gamma)\mathbb{E}_{s\sim\mu_0,a\sim\pi(\cdot\,|\,s)}\left[Q(s,a)\right]$$

$$\text{s.t.} \quad Q(s,a) \geq r(s,a) + \gamma\mathbb{E}_{s'\sim P(\cdot\,|\,s,a),a'\sim\pi(\cdot\,|\,s')}\left[Q(s',a')\right], \quad \forall\,(s,a)\in\mathbb{S}\times\mathbb{A}\,.$$

One can show that $Q_\pi^{P^*}(s,a)$ is the optimal solution of the above (primal) optimization problem. The dual form of this optimization problem can be reformulated as

$$\max_{d:\mathbb{S}\times\mathbb{A}\to\mathbb{R}_+} \quad \mathbb{E}_d[r(s,a)]$$

$$\text{s.t.} \quad d(s',a') = (1-\gamma)\mu_0(s')\pi(a'\,|\,s') + \gamma\sum_{s,a}d(s,a)P(s'\,|\,s,a)\pi(a'\,|\,s'), \quad \forall\,(s',a')\in\mathbb{S}\times\mathbb{A}\,.$$

One can show that the stationary distribution $d_{\pi,\gamma}^{P^*}$ is the solution to the above dual problem.

For the dual problem, we introduce the Lagrange multiplier $Q(s,a)$ for each $(s,a)$ tuple, and the Lagrangian can be written as

$$L(d,Q) := \mathbb{E}_d[r(s,a)] + (1-\gamma)\mathbb{E}_{\mu_0,\pi}[Q(s,a)] + \mathbb{E}_{d,P,\pi}[\gamma Q(s',a') - Q(s,a)]\,.$$

From the previous observations, the *optimal Lagrange multiplier* for this Lagrangian is $Q_\pi^{P^*}(s,a)$, which is the solution to the primal problem. Changing $d$ to the MIW $\omega$, we have

$$L(\omega,Q) = \underbrace{\mathbb{E}_{(s,a)\sim d_{\pi_b,\gamma}^{P^*}}\left[\omega(s,a)r(s,a)\right]}_{\text{①: Estimator for } J(\pi,P^*) \text{ via } \omega(s,a)}$$
$$+ \underbrace{(1-\gamma)\mathbb{E}_{\mu_0,\pi}[Q(s,a)] + \mathbb{E}_{(s,a,s')\sim d_{\pi_b,\gamma}^{P^*},a'\sim\pi(\cdot\,|\,s')}[\gamma\omega(s,a)Q(s',a') - \omega(s,a)Q(s,a)]}_{\text{②: Loss for } \omega(s,a) \text{ that ideally should be 0}}, \quad (22)$$

where the first term ① estimates the expected return $J(\pi,P^*)$ via $\omega(s,a)$, which is accurate if $\omega$ is the true density ratio. The second term ② of Eq. (22) is a loss for $\omega(s,a)$ as in Eq. (21), except that we replace the test function $f(s,a)$ with $Q(s,a)$. With the *optimal Lagrange multiplier* $Q_\pi^{P^*}(s,a)$, we can get rid of the inner minimization of the original maximin problem. In practice, since $Q_\pi^{P^*}$ is unknown, one may directly set $Q$ to be the "estimated optimal multiplier" $Q_\pi^{\widehat{P}}$ and optimize $\omega$ solely.

From Eq. (22), the MIW $\omega$ can be optimized via two alternative approaches. **(1)** We can directly optimize $\omega$ with $L(\omega,Q_\pi^{\widehat{P}})$, whose optimum should be close to $J(\pi,P^*)$. **(2)** We can optimize $\omega$ *w.r.t.* ②, because we know that if ② is close to zero, $\omega$ is close to the true MIW, and the resulting $L(\omega,Q_\pi^{\widehat{P}})$ is again close to $J(\pi,P^*)$ based on the term ①. In theory the main difference of these two approaches is that we may obtain a more accurate estimate for $J(\pi,P^*)$ using approach **(1)** since we optimize $\omega$ on both the first and the second terms. However, since the true reward function $r(s,a)$ is unknown, when we optimize $\omega$ using approach **(1)**, the optimization process may be unstable due to the approximation error for $r(s,a)$. By contrast, for approach **(2)**, we know that $\omega$ is accurate when ② is close to zero, thus we can leverage the MSE loss to optimize $\omega$ so that ② is shrunk towards zero. This approach is similar to the supervised learning and can be more stable in practice. Since our goal is to obtain a good estimate of the MIW $\omega$, not the numerical value of $J(\pi,P^*)$, we therefore choose approach **(2)** that optimize $\omega$ *w.r.t.* ②.

Another intuitive motivation for using $Q_\pi^{P^*}$ as the test function is that the minimax loss Eq. (21) for optimizing $\omega$ is based on a *saddle-point optimization*, and $f \in \mathcal{F}$ is the *dual variable* for the MIW $\omega$. From the previous discussion, MIW and the action-value function have some primal-dual relationship, and $Q_\pi^{P^*}$ is the optimal "dual" variable for the off-policy evaluation problem *w.r.t.* the MIW $\omega$. Since the dual variable $f$ in Eq. (21) is indeed some action-value function [29], we may set $f$ to be this optimal "dual" of $\omega$, which will lead to the choice of $Q_\pi^{P^*}$ as the test function.

**Remark.** Note that both our approach and VPM choose the test function based on some mathematical relationships between the test function and the MIW $\omega$, so that the objective for $\omega$ could potentially be easier to optimize. Both methods are not the *best theoretical way* to optimize $\omega$. The best theoretical way is simply using the saddle-point-optimization based methods such as DualDICE, GenDICE, or a direct minimax optimization on $L(\omega, Q)$ in Eq. (22). However, as we discussed above, there are some practical difficulties for the saddle-point-optimization based approaches. As we observed in Fig. 2, the empirical results indeed show that our approach and VPM are more stable than DualDICE and GenDICE. Further, it is feasible to use other functions as the test function, such as the reward function or any other initialized neural networks. However, these choices may not have nice mathematical properties, such as being the fixed-point solution or some primal-dual relationship, and thus the resulting objective may not be easier to optimize. Table 3 in Appendix A contains an ablation study where we use the reward function as the test function to optimize $\omega$. Indeed, this variant generally performs worse than our main method that uses the Q-function as the test function.

## D   Using value function as the discriminator of model training

Eq. (18) of the proof of Theorem 1 motivates us to use the value function $V_\pi^{\widehat{P}}(\cdot)$ as the discriminator of the model training. Specifically, the model training objective is

$$\arg\min_{\widehat{P} \in \mathcal{P}} \mathbb{E}_{(s,a) \sim d_{\pi_b,\gamma}^{P^*}} \left\{ \omega(s,a) \cdot \left| \mathbb{E}_{s' \sim P^*(\cdot \,|\, s,a)} \left[ V_\pi^{\widehat{P}}(s') \right] - \mathbb{E}_{s' \sim \widehat{P}(\cdot \,|\, s,a)} \left[ V_\pi^{\widehat{P}}(s') \right] \right| \right\}, \quad (23)$$

where the discriminator $V_\pi^{\widehat{P}}$ can be implemented by the current estimate of the action-value function, which is treated as fixed during the model training. We implement this idea as a variant where the model is trained by the weighed objective Eq. (23) that uses the value-function estimate to discriminate the predicted transition from the real. The reward function $\hat{r}$ is still estimated by the weighted-MLE objective. The model is initialized by the MLE loss. Other technical details follow the main algorithm. We dubbed this variant as "V-Dis".

Table 4 compares this variant with our main method. We see that V-Dis underperforms our main method on almost all dataset, often by a large margin. The inferior results of V-Dis may be related to the coupled effect on model training from the inaccurate value-function estimation. In Eq. (23), this inaccuracy can affect both the value difference (the $|\cdot|$ term) and the MIW $\omega$. Our fix of $V_\pi^{\widehat{P}}$ during the model training process may also affect the performance. Future work can investigate how to train $V_\pi^{\widehat{P}}$ together with $\widehat{P}$ in the model estimation.

## E   Technical details

### E.1   Details for the main algorithm

Our main algorithm consists of three major components: model training (Appendix E.1.1), the marginal importance weight training (Appendix E.1.2), and RL policy training via actor-critic algorithm (Appendix E.1.3).

The non-zero assumption of $Q_\pi^{\widehat{P}}$ in the derivation of Eq. (11) motivates us to scale the rewards in the offline dataset to be roughly within $(0, 1]$, similar to Chen et al. [82]. Specifically, before starting our main algorithm, we normalize the rewards $r_i$'s in the offline dataset via $(r_i - r_{\min} + 0.001)/(r_{\max} - r_{\min})$, where $r_{\min}$ and $r_{\max}$ respectively denote the minimum and maximum reward in the offline dataset.

Table 4: Normalized returns for experiments on the D4RL tasks. Mean and standard deviation across five random seeds are reported. "V-Dis" denotes the variant where the model is trained by the value-function-discriminated weighted objective Eq. (23). The results of our main method in Section 4.1 is reported in column Main. Here, "hcheetah" denotes "halfcheetah," "med" denotes "medium," "rep" denotes "replay," and "exp" denotes "expert."

| Task Name | V-Dis | Main |
|---|---|---|
| maze2d-umaze | $36.7 \pm 28.3$ | $55.8 \pm 3.9$ |
| maze2d-med | $36.9 \pm 19.0$ | $107.2 \pm 45.0$ |
| maze2d-large | $23.9 \pm 17.6$ | $180.0 \pm 39.3$ |
| hcheetah-med | $38.3 \pm 17.1$ | $51.7 \pm 0.4$ |
| walker2d-med | $46.9 \pm 28.3$ | $83.1 \pm 1.8$ |
| hopper-med | $58.4 \pm 7.4$ | $58.9 \pm 7.9$ |
| hcheetah-med-rep | $21.3 \pm 13.9$ | $44.6 \pm 0.7$ |
| walker2d-med-rep | $16.3 \pm 12.9$ | $81.5 \pm 3.0$ |
| hopper-med-rep | $69.0 \pm 19.6$ | $91.1 \pm 9.5$ |
| hcheetah-med-exp | $39.7 \pm 4.2$ | $90.2 \pm 2.0$ |
| walker2d-med-exp | $51.3 \pm 29.4$ | $107.7 \pm 2.1$ |
| hopper-med-exp | $24.0 \pm 8.0$ | $76.0 \pm 15.7$ |
| pen-human | $11.4 \pm 10.0$ | $20.6 \pm 10.7$ |
| pen-cloned | $36.3 \pm 10.5$ | $57.4 \pm 19.2$ |
| pen-exp | $131.3 \pm 16.7$ | $138.3 \pm 6.6$ |
| door-exp | $60.4 \pm 33.0$ | $96.3 \pm 9.8$ |
| Average Score | 43.9 | 83.8 |

### E.1.1 Model training

We follow the literature [83, 15, 16, 47] to assume no prior knowledge about the reward function and thus use neural network to approximate transition dynamic and the reward function. Our model is constructed as an ensemble of Gaussian probabilistic networks . Except for the weighted loss function, we use the same model architecture, ensemble size (=7), number of elite model (=5), train-test set split, optimization scheme, elite-model selection criterion, and sampling method as in Yu et al. [16]. To save computation, we define each epoch of model training as 1000 mini-batch gradient steps, instead of the original `n_train / batch_size`.

As in Yu et al. [16], the input to our dynamic model is $\left((s,a) - \mu_{(s,a)}\right)/\sigma_{(s,a)}$, where $\mu_{(s,a)}$ and $\sigma_{(s,a)}$ are respectively coordinate-wise mean and standard deviation of $(s,a)$ in the offline dataset. We follow Kidambi et al. [19] and Matsushima et al. [84] to define the learning target as $\left((r, \Delta s) - \mu_{(r,\Delta s)}\right)/\sigma_{(r,\Delta s)}$, where $\Delta s$ denote $s' - s$, $\mu_{(r,\Delta s)}$ and $\sigma_{(r,\Delta s)}$ denote the coordinate-wise mean and standard deviation of $(r, \Delta s)$ in the offline dataset. As in prior work using Gaussian probabilistic ensemble on model-based RL [83, 15, 16, 21, 18], we use a double-head architecture for our dynamic model, where the two output heads represent the mean and log-standard-deviation of the normal distribution of the predicted output, respectively. We augment the maximum likelihood loss in Yu et al. [16] as the weighted maximum likelihood loss, independently for each model in the ensemble. Specifically, the dynamic model is initially trained using un-weighted maximum likelihood loss. We then employ the warm-start strategy to start each subsequent weighted re-training from the current dynamic model, with the weights $\omega(s,a)$ from the latest marginal importance weight network. To unify the learning rate across each run of model (re-)training, the weights $\omega(s,a)$'s are normalized so that their mean is 1 across the offline dataset.

To improve training stability, we are motivate by Hishinuma and Senda [47] to heuristically augment the environmental termination function by (1) $\text{any}(|\hat{s}'| > 2 \cdot \max_{s,s' \in \mathcal{D}_{\text{env}}} \{|s|, |s'|\})$ where all operations are coordinate-wise, *i.e.*, whether any coordinate of the predicted next state is outside of this relaxed observation-range in the offline dataset; and (2) $|\hat{r}(s,a)| > r_{\text{range}} \triangleq \max(|r_{\min} - 10 \cdot \sigma_r|, |r_{\max} + 10 \cdot \sigma_r|)$ where $r_{\min}, r_{\max}, \sigma_r$ are the minimum, maximum, and standard deviation of rewards in the offline dataset, *i.e.*, whether the predicted reward is outside of this relaxed reward-range in the offline dataset. To penalize the policy for visiting out-of-distribution

state-action pair, we modify $\hat{r}(s,a)$ to be $-r_{\text{range}}$ if either (1) or (2) is violated. No modification for $\hat{r}(s,a)$ is performed if only the environmental termination is triggered.

### E.1.2 Marginal importance weight training

We follow the OPE literature [27, 43, 30] to assume that the initial-state distribution $\mu_0$ is known. In the implementation, we augment $\mathcal{D}_{\text{env}}$ with $10^5$ samples from the ground-truth $\mu_0$. To enhance training stability, we are motivated by the double Q-learning [85] to use a target marginal importance weight network $\omega'(s,a)$ and a target critic network $Q'(s,a)$ in the implementation of MIW training. We employ the warm-start strategy so that each retraining of the marginal importance weight starts from the current $\omega(s,a)$. On each run of the weight-retraining, $Q'$ is initialized as the current critic target $Q_{\boldsymbol{\theta}'}$. Since the weights $\omega(s,a)$'s are normalized to have mean 1 in the weighted re-training of the dynamic model, to narrow the gap between the learning and the application of the weights, we add a constraint into the training of $\omega$ that its mean on each mini-batch is upper bounded by some constant $g_{\text{constraint}}$. Specifically, on each mini-batch gradient step, MIW is trained by the following constraint optimization

$$\arg\min_{\omega} \left( \frac{1}{|\mathcal{B}|} \sum_{(s_i,a_i)\in\mathcal{B}} \omega(s_i,a_i) \cdot Q_{\boldsymbol{\theta}}(s_i,a_i) - y \right)^2 ,$$

$$\text{s.t.} \quad \frac{1}{|\mathcal{B}|} \sum_{(s_i,a_i)\in\mathcal{B}} \omega(s_i,a_i) \leq g_{\text{constraint}}, \tag{24}$$

$$\text{where} \quad y = \gamma \cdot \frac{1}{|\mathcal{B}|} \sum_{\substack{(s_i,a_i,s_i')\in\mathcal{B} \\ a_i' \sim \pi_{\boldsymbol{\phi}}(\cdot\,|\,s_i')}} \omega'(s_i,a_i) \cdot Q'(s_i',a_i') + (1-\gamma) \cdot \frac{1}{|\mathcal{B}_{\text{init}}|} \sum_{\substack{s_0\in\mathcal{B}_{\text{init}} \\ a_0 \sim \pi_{\boldsymbol{\phi}}(\cdot\,|\,s_0)}} Q'(s_0,a_0),$$

where $g_{\text{constraint}}$ is conveniently chosen as 10 and $\mathcal{B}, \mathcal{B}_{\text{init}} \sim \mathcal{D}_{\text{env}}$. To optimize Eq. (24), we use the constraint optimization method in Gong et al. [86] with base optimizer Adam [87]. For training stability, the gradient norm is clipped to be bounded by 1.

As in double Q-learning, we update $Q'$ towards $Q_{\boldsymbol{\theta}}$ and $\omega'$ towards $\omega$ via exponential moving average with rate 0.01 after each gradient step. We use batch sizes $|\mathcal{B}| = 1024$ and $|\mathcal{B}_{\text{init}}| = 2048$ to train the marginal importance weight model $\omega(s,a)$.

### E.1.3 Actor-critic training

Our policy training consists of the following six parts.

**Generate synthetic data using dynamic model.** We follow Janner et al. [15] and Yu et al. [16] to perform $h$-step rollouts branching from the offline dataset $\mathcal{D}_{\text{env}}$, using the learned dynamic $\widehat{P}, \hat{r}$, and the current policy $\pi_{\boldsymbol{\phi}}$. The generated data is added to a separate replay buffer $\mathcal{D}_{\text{model}}$.

To save computation, we generate synthetic data every `rollout_generation_freq` iterations.

**Critic training.** Sample mini-batch $\mathcal{B} \sim \mathcal{D} = f \cdot \mathcal{D}_{\text{env}} + (1-f) \cdot \mathcal{D}_{\text{model}}$, with $f \in [0,1]$. For each $s' \in \mathcal{B}$, sample one corresponding actions $a' \sim \pi_{\boldsymbol{\phi}'}(\cdot\,|\,s')$. Calculate the estimate of the action-value of the next state-action pair as

$$\widetilde{Q}(s,a) \triangleq c \min_{j=1,2} Q_{\boldsymbol{\theta}_j'}(s',a') + (1-c) \max_{j=1,2} Q_{\boldsymbol{\theta}_j'}(s',a').$$

Calculate the critic-learning target defined as

$$\widetilde{Q}(s,a) \leftarrow r(s,a) + \gamma \cdot \widetilde{Q}(s,a) \cdot \left( \left| \widetilde{Q}(s,a) \right| < 2000 \right), \tag{25}$$

where 2000 is conveniently chosen to enhance the termination condition stored in $\mathcal{B}$ for training stability. Then minimize the critic loss with respect to the double critic networks, over $(s,a) \in \mathcal{B}$, with learning rate $\eta_{\boldsymbol{\theta}}$,

$$\forall j = 1, 2, \ \arg\min_{\boldsymbol{\theta}_j} \frac{1}{|\mathcal{B}|} \sum_{(s,a)\in\mathcal{B}} \text{Huber}\left( Q_{\boldsymbol{\theta}_j}(s,a), \widetilde{Q}(s,a) \right),$$

where $\text{Huber}(\cdot)$ denote the Huber-loss with threshold conveniently chosen as 500. Finally, the norm of the gradient for critic update is clipped to be bounded by 0.1 for training stability.

As a remark, the mentioned enhanced termination condition, Huber loss and gradient-norm clipping are engineering designs for stable training in combating for not knowing the true reward function. We use a unified setting for these designs across all tested datasets in all experiments, and thus a per-dataset tuning may further improve our results.

**Discriminator training.** The "fake" samples $\mathcal{B}_{\text{fake}}$ for discriminator training is formed by the following steps: first, sample $|\mathcal{B}|$ states $s \sim \mathcal{D}$; second, for each $s$ get one corresponding action $a \sim \pi_{\boldsymbol{\phi}}(\cdot \,|\, s)$; third, for each $(s, a)$ get an estimate of the next state $s'$ using $\widehat{P}$ as $s' \sim \widehat{P}(\cdot \,|\, s, a)$, with terminal states removed; fourth, sample one next action $a'$ for each next state $s'$ via $a' \sim \pi_{\boldsymbol{\phi}}(\cdot \,|\, s')$. The "fake" samples $\mathcal{B}_{\text{fake}}$ is subsequently formed by

$$\mathcal{B}_{\text{fake}} \triangleq \begin{bmatrix} (s, & a) \\ (s', & a') \end{bmatrix}.$$

Note that terminal states are removed since by definition, no action choices are needed on them. The "true" samples $\mathcal{B}_{\text{true}}$ consists of $|\mathcal{B}_{\text{fake}}|$ state-action samples from $\mathcal{D}_{\text{env}}$.

The discriminator $D_{\boldsymbol{\psi}}$ is optimized with learning rate $\eta_{\boldsymbol{\psi}}$ as

$$\arg\max_{\boldsymbol{\psi}} \frac{1}{|\mathcal{B}_{\text{true}}|} \sum_{(s,a) \sim \mathcal{B}_{\text{true}}} \log D_{\boldsymbol{\psi}}(s, a) + \frac{1}{|\mathcal{B}_{\text{fake}}|} \sum_{(s,a) \sim \mathcal{B}_{\text{fake}}} \log \left(1 - D_{\boldsymbol{\psi}}(s, a)\right).$$

**Actor training.** The policy is updated once every $k$ updates of the critic and the discriminator. Using the "fake" samples $\mathcal{B}_{\text{fake}}$ and the discriminator $D_{\boldsymbol{\psi}}$, the generator loss for actor training is

$$\mathcal{L}_g(\boldsymbol{\phi}) = \frac{1}{|\mathcal{B}_{\text{fake}}|} \sum_{(s,a) \in \mathcal{B}_{\text{fake}}} \left[\log\left(1 - D_{\boldsymbol{\psi}}(s, a)\right)\right].$$

The policy is optimized with learning rate $\eta_{\boldsymbol{\phi}}$ as

$$\arg\min_{\boldsymbol{\phi}} -\lambda \cdot \frac{1}{|\mathcal{B}|} \sum_{s \in \mathcal{B}, a \sim \pi_{\boldsymbol{\phi}}(\cdot \,|\, s)} \left[\min_{j=1,2} Q_{\boldsymbol{\theta}_j}(s, a)\right] + \mathcal{L}_g(\boldsymbol{\phi}),$$

where in all tested datasets the regularization coefficient $\lambda \triangleq 10/Q_{avg}$ with soft-updated $Q_{avg}$ and the penalty coefficient 10 conveniently chosen, similar to Fujimoto and Gu [42].

**Soft updates.** We follow Fujimoto et al. [11] to soft-update the target-network parameter and the $Q_{avg}$ value,

$$\boldsymbol{\phi}' \leftarrow \beta\boldsymbol{\phi} + (1-\beta)\boldsymbol{\phi}',$$
$$\boldsymbol{\theta}'_j \leftarrow \beta\boldsymbol{\theta}_j + (1-\beta)\boldsymbol{\theta}'_j, \quad \forall j = 1, 2,$$
$$Q_{avg} \leftarrow \beta \frac{1}{|\mathcal{B}|} \sum_{s \in \mathcal{B}, a \sim \pi_{\boldsymbol{\phi}}(\cdot \,|\, s)} \left|\min_{j=1,2} Q_{\boldsymbol{\theta}_j}(s, a)\right| + (1-\beta)Q_{avg}.$$

Soft-updates are performed after each iteration.

**Warm-start step.** We follow Kumar et al. [45] and Yue et al. [88] to warm-start our policy training in the first $N_{\text{warm}}$ epochs. In this step, the policy is optimized *w.r.t.* the generator loss $\mathcal{L}_g(\boldsymbol{\phi})$ only, *i.e.*, $\arg\min_{\boldsymbol{\phi}} \mathcal{L}_g(\boldsymbol{\phi})$. The learning rate $\eta_{\boldsymbol{\phi}}$ in the warm-start step is the same as the normal training step.

### E.2 Details for the continuous-control experiment

**Datasets.** We evaluate algorithms on the continuous control tasks in the D4RL benchmark [31]. Due to limited computational resources, we select therein a representative set of datasets. Specifically, **(1)** we select the "medium-expert," "medium-replay," and "medium" datasets for the Hopper, HalfCheetah, and Walker2d tasks in the Gym-MuJoCo domain, which are commonly used benchmarks in prior

work [12, 13, 89, 45]. As in recent literature [21, 90, 91], we do not test on the "random" and "expert" datasets, as they are considered as less practical [84] and can be simply solved by directly applying standard off-policy RL algorithms [14] and behavior cloning on the offline dataset. We note that a data-quality agnostic setting may not align with practical offline RL applications, *i.e.*, one typically should know the quality of the offline datasets, *e.g.*, whether it is collected by random policy or by experts. **(2)** Apart from the Gym-MuJoCo domain, we further consider the Maze2D domain of tasks[2] for the non-Markovian data-collecting policy and the Adroit tasks[3] [92] for their high dimensionality and sparse rewards.

**Evaluation protocol.** Our algorithm is trained for 1000 epochs, where each epoch consists of 1000 mini-batch gradient descent steps. After each epoch of training, a rollout of 10 episodes is performed. Both our algorithm and baselines are run under five random seeds $\{0, 1, 2, 3, 4\}$, and we report the mean and standard deviation of the final rollouts across the five seeds. We follow Fu et al. [31] to rerun the baselines under the recommended hyperparameter setting, including dataset-specific hyperparameters if available.

**Results of CQL.** We run CQL using the implementation suggested by Kostrikov et al. [91], which only provide hyperparameters for the Gym-MuJoCo and AntMaze domains. For the Maze2D and Adroit datasets, we run both sets of recommended hyperparameters and per-dataset select the better result to report. We note that hyperparameter settings for the Maze2D and Adroit datasets are also not provided in the original CQL implementation. An in-depth tuning of CQL on these two domains of datasets is beyond the scope of this paper.

**Implicit policy implementation.** We follow White [93] to choose the noise distribution as the isotropic normal, *i.e.*, $p_z(z) \triangleq \mathcal{N}\left(\mathbf{0}, \sigma_{\mathrm{noise}}^2 \boldsymbol{I}\right)$, where the default setting of the dimension of $z$ is $\dim(z) = \mathtt{min(10, state\_dim//2)}$ and of the noise standard deviation is $\sigma_{\mathrm{noise}} = 1$. In the forward pass of the implicit policy, for a given state $s$, a noise sample $z \sim p_z(z)$ is first independently drawn. Then $s$ is concatenated with $z$ and the resulting $[s, z]$ is inputted into the (deterministic) policy network to sample stochastic action.

**Terminal states.** In practice, the episodes in the offline dataset have finite horizon. Thus, special treatment is needed for those terminal states in calculating the Bellman update target. We combine common practice [94, 95] with our critic update target Eq. (25) as

$$\widetilde{Q}(s, a) = \begin{cases} r(s, a) + \gamma \cdot \widetilde{Q}(s, a) \cdot \left(\left|\widetilde{Q}(s, a)\right| < 2000\right) & \text{if } s' \text{ is a non-terminal state} \\ r(s, a) & \text{if } s' \text{ is a terminal state} \end{cases},$$

since the action-value function is undefined on the terminal $s'$ due to no action choice therein.

Our network architecture is presented in Appendix E.2.1. To save computation, we use simple neural networks as in Fujimoto et al. [12]. A fine-tuning of the noise distribution $p_z(z)$, the network architecture, the optimizer hyperparameters, *etc.*, is left for future work.

### E.2.1 Details for the implementation of AMPL

With the training techniques for GAN suggested by previous work in generative modeling, we are able to stably and effectively train the GAN structure on data with moderate dimension, *e.g.*, the D4RL datasets we consider. In this paper, we adopt the following techniques from the literature.

- Following Goodfellow et al. [35], $\pi_\phi$ is trained to maximize $\mathbb{E}_{(s,a)\in\mathcal{B}_{\mathrm{fake}}}\left[\log\left(D_\psi(s, a)\right)\right]$.

- We are motivated by Radford et al. [96] to use LeakyReLU activation function in both the generator and the discriminator, with $\mathtt{negative\_slope}$ being the default value 0.01.

- For stable training, we follow Radford et al. [96] to use a reduced momentum term $\beta_1 = 0.4$ in the Adam optimizers of the policy and the discriminator networks; and to use learning rates $\eta_\phi = \eta_\psi = 2 \times 10^{-4}$.

---

[2]We use the tasks "maze2d-umaze," "maze2d-medium," and "maze2d-large."
[3]We use the tasks "pen-human," "pen-cloned," "pen-expert," and "door-expert."

- To avoid discriminator overfitting, we are motivated by Salimans et al. [97] and Goodfellow [98] to use one-sided label smoothing with soft and noisy labels. Specifically, random numbers in $[0.8, 1.0)$ are used as the labels for the "true" sample $\mathcal{B}_{\text{true}}$. No label smoothing is needed for the "fake" sample $\mathcal{B}_{\text{fake}}$, *i.e.*, their labels are all 0.

- Binary cross entropy loss between the labels and the discriminator outputs is used as the loss for the discriminator training in the underlying GAN structure.

- We are motivated by TD3 [11] and GAN to update the policy $\pi_\phi(\cdot \,|\, s)$ once per $k$ updates of the critics and the discriminator.

Table 5 shows the shared hyperparameters for all datasets in our experiments. *Many of these hyperparameters follow the literature without further tuning.* For example, we use $\eta_\phi = \eta_\psi = 2 \times 10^{-4}$ as in Radford et al. [96], $\eta_\theta = 3 \times 10^{-4}$ and $N_{\text{warm}} = 40$ as in Kumar et al. [45], $c = 0.75$ as in Fujimoto et al. [12], policy frequency $k = 2$ as in Fujimoto et al. [11], $f = 0.5$ as in Yu et al. [18] and model learning rate $\eta_{\widehat{P}} = 0.001$ as in Yu et al. [16]. Unless specified, the shared and the non-shared hyperparameters are used throughout the main results and the ablation study.

Table 5: Shared hyperparameters for AMPL.

| Hyperparameter | Value |
|---|---|
| Optimizer | Adam |
| Training iterations | $10^6$ |
| Training iterations for $\omega(s, a)$ | $10^5$ |
| Model retrain period | per 100 epochs |
| Learning rate $\eta_\theta$ | $3 \times 10^{-4}$ |
| Learning rate $\eta_\phi, \eta_\psi$ | $2 \times 10^{-4}$ |
| Learning rate $\eta_{\widehat{P}}$ | $1 \times 10^{-3}$ |
| Learning rate $\eta_\omega$ for $\omega(s, a)$ | $1 \times 10^{-6}$ |
| Penalty coefficient | 10 |
| Batch size | 512 (as in Lee et al. [44]) |
| Discount factor $\gamma$ | 0.99 |
| Target network update rate $\beta$ | 0.005 |
| Weighting for clipped double Q-learning $c$ | 0.75 |
| Noise distribution $p_z(z)$ | $\mathcal{N}\left(\mathbf{0}, \sigma_{\text{noise}}^2 \boldsymbol{I}\right)$ |
| Default value of $\sigma_{\text{noise}}$ | 1 |
| Policy frequency $k$ | 2 |
| Rollout generation frequency | per 250 iterations |
| Number of model-rollout samples per iteration | 128 |
| Rollout retain epochs | 5 |
| Real data percentage $f$ | 0.5 |
| Warm start epochs $N_{\text{warm}}$ | 40 |
| Random seeds | $\{0, 1, 2, 3, 4\}$ |

Since the tested datasets possess diverse nature, we follow the common practice in offline model-based RL to perform gentle dataset-specific hyperparameter tuning. Details are discussed below.

We are motivated by Janner et al. [15] and Yu et al. [16] to consider the rollout horizon $h \in \{1, 3, 5\}$. We use $h = 1$ for halfcheetah-medium, hopper-medium, hopper-medium-replay, maze2d-large, pen-expert, door-expert, pen-human; $h = 3$ for hopper-medium-expert, walker2d-medium, halfcheetah-medium-replay, maze2d-umaze, maze2d-medium; and $h = 5$ for halfcheetah-medium-expert, walker2d-medium-expert, walker2d-medium-replay, pen-cloned.

Zheng and Zhou [99] and Zhang et al. [100] show that a larger noise dimension, *e.g.*, 50, can help learning a more flexible distribution, while a smaller noise dimension, *e.g.*, 1, can make the distribution leaning towards deterministic. Hence we use the default noise dimension for the tasks: maze2d-umaze, door-expert; noise dimension 50 for the tasks: hopper-medium-expert, halfcheetah-medium-expert, walker2d-medium-expert, halfcheetah-medium, hopper-medium, walker2d-medium, halfcheetah-medium-replay, hopper-medium-replay, walker2d-medium-replay, maze2d-medium, maze2d-large, pen-expert; and noise dimension 1 for the tasks: pen-cloned, pen-human.

Due to the narrow data distribution, we use $\sigma_{\text{noise}} = 0.001$ for datasets pen-human and pen-cloned to avoid training divergence.

We state below the network architectures of actor, critic, discriminator, and MIW in the implementation of AMPL. Motivated by the clipped double Q-learning, two critic networks are used with the same architecture.

Actor

Linear(state_dim + noise_dim, 400)
LeakyReLU
Linear(400, 300)
LeakyReLU
Linear(300, action_dim)
max_action × tanh

Critic

Linear(state_dim + action_dim, 400)
LeakyReLU
Linear(400, 300)
LeakyReLU
Linear(300, 1)

Discriminator

Linear(state_dim + action_dim, 400)
LeakyReLU
Linear(400, 300)
LeakyReLU
Linear(300, 1)
Sigmoid

MIW

Linear(state_dim + action_dim, 400)
LeakyReLU
Linear(400, 300)
LeakyReLU
Linear(300, 1)
$(\text{softplus}(\cdot - 10^{-8}) + 10^{-8}).\text{power}(\alpha)$

In the MIW network, $\alpha$ is the smoothing exponent to smooth the distribution of the weight estimates in the offline dataset, whose design is motivated by Hishinuma and Senda [47]. We fix $\alpha$ as $0.5$ on the MuJoCo datasets and as $0.2$ on the Maze2D and Adroit datasets. The output of the MIW network prior to $\text{power}(\alpha)$ is lower bounded by $10^{-8}$ to avoid gradient explode. We follow Dai et al. [69] to initialize the weights of last layer of the MIW network from $\text{Uniform}(-0.003, 0.003)$, and bias as $0$.

### E.3 Details for the implementation of alternative MIW estimation methods in Section 4.2

We implement and train the alternative methods VPM, GenDICE, and DualDICE in Section 4.2 based on the corresponding papers and source codes. To stabilize training, the design of target MIW network in Section E.1.2 is applied wherever applicable. The target MIW network is hard-updated, as suggested by the corresponding papers. The $\nu$ network in GenDICE and DualDICE has the same architecture as the critic network and has its own hard-updated target network. We follow the official implementation of GenDICE and DualDICE to clip the gradient values by $1$. For the convex function $f$ in DualDICE, we use $f(x) = \frac{2}{3}|x|^{3/2}$, as suggested in its Section 5.3.

## F  Potential negative societal impacts

Since offline RL has strong connection with the supervised learning, offline RL methods, including the proposed AMPL in this paper, are subject to the bias in the offline dataset. The data-bias can be exploited by both the dynamic-model learning and the policy training.

## G  Computing resources

We ran all experiments on 4 NVIDIA Quadro RTX 5000 GPUs.