# OpenReview forum: "A Unified Framework for Alternating Offline Model Training and Policy Learning"
_NeurIPS.cc/2022/Conference — NeurIPS 2022 Accept_

### Official Review · Reviewer_A7rq · 2022-07-10

**Rating:** 7
**Confidence:** 4
**Soundness:** 3 good
**Presentation:** 3 good
**Contribution:** 3 good

**Summary:**

In this work, the authors investigate a new approach to the problem of Model-Based Reinforcement Learning in the off-line RL setting.
The paper proposes a number of new ideas. First, instead of first learning a model and subsequently using it to obtain a policy, their approach alternates between updating the model and improving the policy. The MLE model learning is aware of the discrepancy between the stationary distribution of the behavior policy and of the learned one through the use of marginal importance weighting. These weights are learned using a novel Bellman backup operator. At the same time the authors propose a new regularization loss in their policy updates to account for the difference between the true transition probabilities and the ones produced by the learned model. This encourages the imaginary rollouts produced by the model and policy to stay close to the true distribution observed in the offline dataset.


**Questions:**

The authors should look at the work proposed in [1] as another simpler method where model and policy learning are coupled to achieve state-of-the-art performance. Is there any benefit in using transition models instead of value equivalent ones ?

Minor suggestions:
Cite and clarify the idea of target networks.
Cite and clarify the use of a Huber loss in the Q-function update rule.

[1] Schrittwieser J, Hubert TK, Mandhane A, Barekatain M, Antonoglou I, Silver D (2021) Online and Offline Reinforcement Learning by Planning with a Learned Model. In: Beygelzimer A, Dauphin Y, Liang P, Vaughan JW (eds) Advances in Neural Information Processing Systems

**Limitations:**

The authors have adequately addressed the limitations and potential negative societal impact of their work.

**Strengths And Weaknesses:**

The authors propose a theoretically and empirically principled algorithm. Their method introduces a number of novel ideas, in a clear way, which are in the right direction of coupling model and policy learning in offline RL.

A possible weakness would be the stability of the learning rules.
The model learning updates depend on the values of the importance weights. At the same time the importance weights are learned through a Bellman backup which depends on the Q values learned using the model. Moreover, the policy is regularized through a GAN mechanism which can further destabilize learning. The authors show empirically that their method is stable which removes some of these concerns. However, since there are many hyperparameters involved it is not clear how important these are in achieving stability.

---

> ### Author Response · Authors · 2022-08-02
> **Response to Reviewer A7rq (Part 1)**
>
> Thank you for your insightful comments. We provide a point-by-point response below.
>
> > **Q1:**  Concern on the stability of the learning rules. The GAN mechanism in the policy regularization can further destabilize learning. Since there are many hyperparameters involved it is not clear how important these are in achieving stability.
>
>
> **A:**
> We note that even though there are several components in our proposed framework. They all contribute to the performance and stability of our Main method that includes all of them.
>
> As discussed in Section 4.2 (a), removing proposed MIW-weighted model (re)training scheme generally worsen the performance. Indeed, as shown on Table 2 in Appendix A, compared with the Main method, the resulting NW variant has a larger standard deviation relative to the mean-score on a large portion of the tested datasets.
>
> As shown on Appendix H of the revision, changing the implicit policy in the Main method to the Gaussian policy again not just leads to worse performance, but also larger standard deviation relative to the mean-score in general. The performance and stability differences are especially significant on the more challenging Maze2D and Adroit datasets [1], which may be related to the insufficiency of the uni-model Gaussian policy in capturing the necessary multiple action modes in these harder datasets compared with an implicit policy, as discussed by [2].
>
> As shown on Appendix H of the revision, removing the proposed regularization term breaks the policy learning, which shows the effectiveness of the proposed regularizer
>
> Appendix H of the revision also shows that removing rollout data in the policy learning again decreases the performance while increasing the instability. Adding model-rollout data can mitigate the off-policy issue in the policy learning by taking into account the rollouts of the learning policy.
>
> Apart from these, we have made efforts to stabilize the training of each component. For example, we use a soft-updated target network $\omega’$ in the training of MIW (Section 3.3), use a conservative target and the Huber loss for the critic training (Section 3.4), etc. We also change the KL divergence in Theorem 1 to the Jensen–Shannon divergence to avoid the practically unstable estimation of the KL dual (Section 3.2). Further, as discussed in Section 3.2, our framework directly utilize many stabilization techniques developed in the GAN community, without tuning the GAN structure much by ourselves.
>
> We also added plots in Appendix G of the revision that shows the relative stability of our proposed MIW estimation method compared to the alternative methods in Section 4.2 (b) (VPM, GenDICE, DualDICE). We refer to Appendix G of the revision for a more detailed discussion. These plots show that our method can still perform well even when the current policy is far away from the behavior policy.
>
> With these considerations, the Main algorithm is quite stable and does not need much hyperparameter tuning.
>
>
> \
> [1] Fu, Justin, et al. "D4rl: Datasets for deep data-driven reinforcement learning." arXiv preprint arXiv:2004.07219 (2020).
> \
> [2] Yang, Shentao, et al. "A Regularized Implicit Policy for Offline Reinforcement Learning." arXiv preprint arXiv:2202.09673 (2022).

---

> > ### Author Response · Authors · 2022-08-02
> > **Response to Reviewer A7rq (Part 2)**
> >
> > > **Q2:** Compare the proposed method with [3].
> >
> > **A:**
> > We thank the reviewer for pointing out the related work.
> >
> > [3] uses a recurrent latent-space model to map the history $h_{1:t}$ till timestep $t$ into an embedding $s_t$, and uses this latent-space model to facilitate the search over future action-sequences in the MCTS method.
> > Because of this latent-space model and the MCTS planning, [3] requires a significant amount of computation on even the feature-based observations, as discussed in Appendix H of [3]. By contrast, we use short-trajectory rollouts and construct the regularizer by only looking-ahead one step. These designs mitigate the compounded error of model rollout without needing a latent-space model — retains stability while saving computation.
> >
> > [3] trains the representation, dynamics and prediction functions jointly, similar to [4], while our alternating model-training and policy-learning scheme can be more interpretable and easier to tune.
> >
> > As discussed in Section 6 of [3], the extension of this MCTS-type method to continuous action space requires some careful consideration. By contrast, as shown by our empirical results, our method naturally works with continuous action space.
> >
> > Finally, we note that our weighted (re-)training of dynamic models can potentially be combined with [3]. As noted by Section 4 of [3], the learned policy by MCTS will increasingly differ from the data-collecting policy. This bias in the state distribution may harm the reward-function learning in [4]. Our method offers a mitigation to such a bias through the incorporation of the marginal-importance-weight in the dynamic-model learning, which may provide a better estimate of $(r, s’)$ on the state-distribution of the learned policy.
> >
> >
> > \
> > [3] Schrittwieser J, Hubert TK, Mandhane A, Barekatain M, Antonoglou I, Silver D (2021) Online and Offline Reinforcement Learning by Planning with a Learned Model. In: Beygelzimer A, Dauphin Y, Liang P, Vaughan JW (eds) Advances in Neural Information Processing Systems
> > \
> > [4] Schrittwieser, Julian, et al. "Mastering atari, go, chess and shogi by planning with a learned model." Nature 588.7839 (2020): 604-609.
> >
> >
> > > **Q3:** Is there any benefit in using transition models instead of value equivalent ones ?
> >
> > **A:** We are not sure if we understand your question. Would you mind explaining it more?
> >
> >
> >
> > > **Q4:** Cite and clarify the idea of target networks and the use of a Huber loss in the Q-function update rule.
> >
> > **A:**
> > Thank you for this suggestion. We have adopted the suggestion in the revised version.

---

> > > ### Author Response · Authors · 2022-08-08
> > > **Following up with Reviewer A7rq**
> > >
> > > Dear Reviewer A7rq,
> > >
> > > Thank you for your appreciation of our work!
> > >
> > > May we kindly ask if our response has addressed your concerns on the stability of our method and on the comparison of our work with the *MuZero Unplugged* paper?
> > >
> > > Since the discussion period is approaching the end, we would like to take the last chance to address any of your remaining/new questions and possibly further improve our manuscript.

---

### Official Review · Reviewer_UQCr · 2022-07-11

**Rating:** 6
**Confidence:** 4
**Soundness:** 2 fair
**Presentation:** 3 good
**Contribution:** 2 fair

**Summary:**



The authors propose a unified framework for alternating model training and policy learning in the offline RL setting. The model and the policy are trained to maximize a lower bound of the actual return. The experiments are conducted in the D4RL benchmark and several SOTA baselines are taken for comparison.

**Questions:**


1. In Section 3.2, to learn a policy, the authors optimize a surrogate objective of Eq.(7) via estimate D_\pi. However, If we have an approximation of D_\pi, why don’t we optimize the model through D_\pi too? Similarly, If we have an approximation of \omega, why don’t we use \omega to estimate D_pi directly for policy optimization?

2. The results in Figure 2 seem great but the details of experiment settings are still unclear to me. Can you give a detailed description (which can be in the Appendix)?  For example, how to normalize the MIWs; What’s the difference among the 1~9 policies for testing.

3. Can the authors provide more evidence to demonstrate the mechanism of the regularizer and MIW training techniques such as conducting experiments in some interpretable environments or recording some intermediate  metrics in some specially designed experiment like (b) in Section 4.2

**Ethics Review Area:**

["I don’t know"]

**Limitations:**


I think the overall idea of the article is interesting but needs some refinements before publication. The authors may be able to further analyze from a theoretical perspective; Or, from an experimental perspective, I think the mechanism of regularization and omega estimation can be more fully experimentally verified. I will consider increasing the score if the authors improve the article in either of the aspect.


**Strengths And Weaknesses:**


Strengths:

1. The motivation for building a unified framework for alternating model training and policy learning is in time and reasonable. Although current offline model-based RL algorithms have learned policies considering the inaccuracy of the models, the objective of model learning is still separated from the offline model-based RL pipeline. It is valuable as a research topic to investigate;

2. The article is overall well-written and easy-to-follow to me;

3. The experiment results in the benchmark datasets seem great.

Weaknesses:

With respect to the theoretical modeling,  there still have large gaps between Theorem 1 and the algorithm designs in Sec. 3.2, 3.3, and 3.4. And thus, I prefer to evaluate the value of the article with respect to the novelty of the proposed techniques.

In Section 3.2 and 3.2, the authors proposed a new regularizer for policy learning and a new technique for importance weight estimation.  Both of the techniques are interesting: the regularizer not only evaluates the difference between two policies but the effect difference of the policies in different models, which makes sense for formulating a unified training framework. Importance weight estimation is also a challenging task. The proposed method utilize a test function to identify and correct the weight estimation. However, there are not enough experiments and descriptions to demonstrate the actual mechanism and the relation of the proposed techniques (See details in the following).

---

> ### Author Response · Authors · 2022-08-02
> **Response to Reviewer UQCr**
>
> Thank you for your thoughtful comments. We provide a point-by-point response below.
>
> > **Q1:** Gaps between Theorem 1 and the algorithm designs in Sec. 3.2, 3.3, and 3.4.
>
> **A:**
> We disagree with the statement that the reviewer made. Our practical algorithm designs still follow the two principles indicated by Theorem 1(L98-L104). The practical changes we made, such as changing from KL to JSD, does not change much the algorithm design indicated by Theorem 1 because both KL and JSD are valid statistical distances for distributions. The main reason for this change is that the KL dual tends to be hard to estimate/optimize in practice.
>
>
> > **Q2:** Why don’t we optimize the model through $D_\pi$ too?
>
> **A:**
> We do optimize the model through $D_\pi$. As discussed in Section 3.1, the weighted-MLE training objective Eq. (6) comes from a direct expansion of the KL term in $D_\pi$, with the terms constant with respect to $\hat P$ removed.
>
> > **Q3:** Given an approximation of $\omega$, why don’t we use $\omega$ to estimate $D_\pi$ directly for policy optimization?
>
> **A:**
> As discussed in Line 133-135, and empirically studied in Section 4.2, we remove the MIW $\omega(s,a)$ during the policy training since we do not observe its empirical benefits. Specifically, incorporating the MIWs into the regularization term may bring in additional instability into the minimax optimization of the policy and the discriminator.
>
> In Appendix H of the revision, we conduct additional ablation studies on changing the JSD regularizer in our Main method to the weighted KL-dual regularizer $D_\pi$. From Table 5 of the revision, it is clear that changing the proposed JSD regularizer in Section 3.2 to the KL-dual based regularizer breaks the policy learning, which may be related to the unstable estimation of KL-dual discussed above and in Section 3.2 of the paper.
>
> > **Q4:** Details on the experimental settings of Figure 2. How to normalize the MIWs? What’s the difference among the 1~9 policies for testing?
>
>
> **A:**
> In general, we implement and train the alternative MIW estimators VPM, GenDICE, and DualDICE in Section 4.2 (b) based on the respective papers and source codes.
>
> As discussed in Section 4.2 (b), for numerical stability, the estimated MIW from these three alternative methods is clipped into $(10^{-8}, 500)$. This is implemented by adding differentiable softplus activations onto the output layer of the MIW network.
>
> As discussed in the caption of Figure 2, we plot the normalized MIWs whose mean on the entire dataset is 1. This is implemented by the code `w = w / w.mean(axis=0)`, where `w` is a tensor for the MIWs of the observations in the offline dataset. This normalization follows prior work [1].
>
> We clarify the confusion on the meaning of the $x$-axis in Figure 2 as follows. Figure 2 shows the boxplots of the MIWs of the observations in the entire dataset. We compare the MIWs from our method (“Main”) and the three alternatives in Section 4.2 (b) during the training process on the “walker2d-medium-replay” dataset. Since the MIW is defined on the learning policy $\pi_\phi$, we retrain the MIW network $\omega$ every 100 epochs during the training process to catch up with the learning policy. The original $x$-axis in Figure 2 represents the retraining time $1-9$, where $1$ is the first retraining of the MIW network after initialization, and $9$ is the last retraining. Equivalently, the $x$-axis can label the epoch count $\\{100, 200, . . . , 900\\}$, on which the retraining is conducted. The y-axis represents the MIW values.
>
> We have changed the $x$-axis of Figure 2 to label the epoch count and the caption accordingly in the revision to clarify this confusion.
>
> \
> [1] Hishinuma, Toru, and Kei Senda. "Weighted model estimation for offline model-based reinforcement learning." Advances in Neural Information Processing Systems 34 (2021): 17789-17800.
>
> > **Q5:** Provide more evidence to demonstrate the mechanism of the regularizer and MIW training techniques
>
> **A:**
> For the regularizer, we include an ablation study in Appendix H (Table 5) that uses KL as the statistical distance, and demonstrate the reason why we change the KL regularizer to JSD. More details can be found in Appendix H.
>
> For the MIW training techniques, we further plot the distribution plots of the $\log(\text{MIW})$ of the entire dataset generated by our method, and by the variants with the three alternative MIW estimation methods in Section 4.2 (b) (VPM, GenDICE, DualDICE) in Figure 3 (Appendix G) of the revision.  In Figure 3, we note that the MIW obtained by these four methods have very different scales, which indicates the instability of the alternative methods compared with ours.
> We refer to Appendix G of the revision for a more detailed discussion on these distribution plots. To conclude, these plots show that our method can still perform well even when the current policy is far away from the behavior policy.

---

> > ### Author Response · Authors · 2022-08-07
> > **Following up with Reviewer UQCr**
> >
> > Dear Reviewer UQCr,
> >
> > Thank you for your careful review!
> >
> > We would like to double check if our response can address your concerns on the gaps between our Theorem 1 and the algorithmic designs, and other related concerns in your review?
> >
> > Please kindly let us know if you have any remaining questions or any further concerns so that we can address them and further revise our manuscript during the discussion period.
> > If our response and revised manuscript have addressed your concerns, please could you re-evaluate our work based on the updated information?

---

> > > ### Comment · Reviewer_UQCr · 2022-08-07
> > > **Response**
> > >
> > > Thanks for your elaborate response and the added experiments.  I have no further concerns.

---

> > > > ### Author Response · Authors · 2022-08-08
> > > > **Thank you for the positive feedback**
> > > >
> > > > Dear Reviewer UQCr,
> > > >
> > > > We are honored that our responses clear your concerns. We deeply appreciate your recommendation of accepting our work.

---

### Official Review · Reviewer_dzbb · 2022-07-11

**Rating:** 6
**Confidence:** 4
**Soundness:** 3 good
**Presentation:** 4 excellent
**Contribution:** 3 good

**Summary:**

The paper focuses on the problem of offline reinforcement learning, where the authors propose a novel model based algorithm. Specifically, the authors start from the general framework of bounding the difference between the return on the true model and a learned model, and then derive the objective for model learning and policy improvement. For the learning dynamics model, the authors propose a maximum likelihood objective weighted by the importance ratio of the policy state action visitation distribution and the dataset distribution. Specifically, the authors introduce a Bellman equation-like objective to estimate this important ratio. For policy improvement, the authors adopt an actor-critic style algorithm and penalizes the divergence between the policy and the dataset during actor updates. The divergence value is estimated using a generative adversarial network.

The authors evaluate the proposed method in the D4RL benchmark, and show that the proposed method outperforms prior model based and model free methods.


**Questions:**

Prior work [1] suggests that some baseline methods used in this paper can be tuned much better. I wonder if the authors could include these improvements in the baseline comparison as well as incorporate them into the proposed method if applicable.


References

[1] Lu, Cong, et al. "Revisiting design choices in offline model based reinforcement learning." International Conference on Learning Representations. 2021.


**Limitations:**

The authors have sufficiently addressed the limitations and societal impacts.


**Strengths And Weaknesses:**

Overall I think the paper presents an interesting idea of doing model based RL from offline dataset. However I do have some concerns.

Pros

The empirical results of the paper are pretty strong. From table 1, it seems that the proposed method achieves better performance than pior method in most of the tasks from D4RL, which is a clear indication that the proposed method provides performance benefits. I also find the ablation study quite useful in showing that the MIW weighted model learning is important.

The paper is well written and the presentation is very clear. Building on top of the general framework of model based RL, the derivations and analysis of the proposed method are intuitive and easy to follow.


Cons

It seems to me that the practical implementation of the algorithm is somewhat disconnected from the derivation of the algorithm, so it is hard to tell if the method works because of the reasons justified by theoretical analysis or because of the choices made in practical implementation. For example, the authors start from the standard approach to bound the gap between the return between the true and learned dynamics model (theorem 1) and propose to estimate the importance ratio (MIW). However, the derived method for estimating MIW is not implementable since it uses quantities not available to the algorithm, so the authors propose to replace the unknown density with the learned Q function. It is not clear to me why this is a good choice and hence I believe that more theoretical analysis is necessary.


During policy training, the authors used a GAN to estimate the divergence, which is one of many ways to estimate the divergence or quantify the uncertainty. Prior work [1] suggests that the performance difference of model based offline RL algorithms mainly come from these practical choices. Hence I believe the paper could benefit from a comparison of different ways of penalizing the policy during policy improvement.


While the paper presents an interesting idea and the empirical results are strong, I do believe that some more analysis and justification is needed for the proposed method. Therefore, I’m neutral between acceptance and rejection for this paper. I highly encourage the authors to include them during the author response period.



References

[1] Lu, Cong, et al. "Revisiting design choices in offline model based reinforcement learning." International Conference on Learning Representations. 2021.


----
The authors' response addressed most of my concerns about the paper and therefore I'm recommending acceptance. I've increased my rating from 5 to 6.

---

> ### Author Response · Authors · 2022-08-02
> **Response to Reviewer dzbb (Part 1)**
>
> Thank you for your detailed feedback. We provide a point-by-point response below.
>
> > **Q1:** The practical implementation is disconnected from the derivation of the algorithm.
>
> **A:**
> We argue that the practical implementations still follow the principles indicated by the theorem.
>
> The two most important principles we discussed in the paper (Line 98-104) have been implemented into the practical algorithms, and have been shown as effective in our empirical experiments.
>
> Further, we conducted additional ablation in Appendix H of the revision, where we removed the components *(i)* the proposed MIW-weighted model (re)training scheme, *(ii)* the proposed regularization term, *(iii)* the rollout data in the policy learning, and *(iv)* change the implicit policy in the Main method to the Gaussian policy. As shown in Table 5 of the revision, removing or changing these components in the Main method  not just leads to worse performance, but also larger standard deviation relative to the mean-score in general
>
> Besides, the main changes that are different from the theorem is that we change KL from JSD. Theoretically there is not much difference between these statistical distances (e.g., KL, JSD, $f$-divergence, etc.), but the practical performance may differ when implemented, since the difficulty of the associated optimization problem varies. Specifically, we clarify the reason why we change from KL to JSD in L125-L132, and further conduct additional ablation study in Appendix H of the revision, where we change the proposed JSD regularizer *(i)* to the KL term in Theorem 1 and *(ii)* to the weighted KL term $D_{\pi}(P^*, \widehat P)$ in Theorem 1. It is clear from Table 5 of the revision that the overall performance of these two variants dropped significantly, which may be related to the practically unstable estimation of the KL dual (Section 3.2)
>
> > **Q2:** The authors propose to replace the unknown density with the learned Q function. Why is this a good choice?  More theoretical analysis is necessary.
>
> **A:**
> We do not replace the unknown density with the learned Q function. What we propose in Section 3.3 is to use the learned Q function as the “test function”, inspired from the primal-dual relationship between Q function and density function when we consider the off-policy evaluation problem as a constraint optimization problem. When we consider the lagrangian function of the constraint optimization problem, we can view Q as the lagrangian multiplier function for the density function, and the action-value function of policy $\pi$ is the optimal multiplier for optimizing the density ratio. In this way we can directly use the learned Q function as the test function. We clarify this in detail in Appendix I.
>
> > **Q3:** The paper could benefit from a comparison of different ways of penalizing the policy during policy improvement.
>
> **A:**
> Thank you for this suggestion.
>
> Apart from the ablation study in Section 4.2, In appendix H of the revision, we provide additional ablation studies on several building blocks of our main method.
>
> To compare different ways of penalizing the policy during the policy improvement, we
> 1. change the JSD regularizer in Main to the $\mathrm{KL}(P^*(s' | s,a) \pi_b(a'| s') || \widehat P(s' | s,a)\pi(a'| s') )$ term in Theorem 1, where $(s,a)$ is sampled from the offline dataset and there is no weighting;
> 2. change the JSD regularizer in Main to the weighted KL term $D_{\pi}(P^*, \widehat P)$ in Theorem 1, with the weighting.
>
> Further, we also consider
>
> 3. changing the implicit policy in our Main method to the Gaussian policy, which is less flexible than the implicit policy;
> 4. removing the proposed regularizer in the policy learning.
>
> From Table 5 in the revision, changing the proposed JSD regularizer to either KL-dual based regularizer breaks the policy learning. This can be related to the practical difficulty in the sample-based estimate of the dual form of the KL divergence, as discussed in Section 3.2 (Line 128-129).
>
> From Table 5 in the revision, changing the implicit policy in our main method to the Gaussian policy generally leads to worse performance, especially on the more challenging Maze2D and Adroit datasets [1]. As discussed by [2], a uni-model Gaussian policy may be insufficient to capture the necessary multiple action modes in these harder datasets, while an implicit policy can be capable.
>
> Finally, removing the proposed regularization term breaks the policy learning, which shows the effectiveness of the proposed regularizer
>
> We agree with the reviewer that the design choices are important in offline model-based RL. We will investigate further other practical choices in our proposed framework in our future work.
>
> \
> [1] Fu, Justin, et al. "D4rl: Datasets for deep data-driven reinforcement learning." arXiv preprint arXiv:2004.07219 (2020).
> \
> [2] Yang, Shentao, et al. "A Regularized Implicit Policy for Offline Reinforcement Learning." arXiv preprint arXiv:2202.09673 (2022).

---

> > ### Author Response · Authors · 2022-08-02
> > **Response to Reviewer dzbb (Part 2)**
> >
> > > **Q4:** Some more analysis and justification is needed for the proposed method.
> >
> > **A:**
> > Thanks for the great suggestions. We have further conducted some ablation to justify the effectiveness of our proposed method. Specifically, we have conducted the following experiments to justify our framework:
> > In Table 5 in Appendix H, we remove the components of MIW, regularization and model rollouts respectively. And we can see that when removing each component proposed in our framework, the performance drops compared with our Main algorithm, which shows the necessity of each component in our framework.
> > We use KL divergence as the original theorem suggested, and the empirical result of using KL divergence is not as good as that of using JSD. We believe this is due to the practical difficulty in the implementation for using KL.
> > We plot the MIW distributions for different MIW training methods (GenDICE, DualDICE, VPM and ours) in Figure 3. As we can see, our proposed method indeed produces stable MIW estimates, showing the efficacy of the proposed MIW training method, so that the MIW can be used for model training effectively.
> >
> > We do agree that we need more rigorous theoretical analysis of the model and policy training framework, which we consider as a future work.
> >
> >
> >
> > > **Q5:** Can the authors include the improvements in [3] in the baseline comparison and incorporate them into the proposed method?
> >
> > **A:**
> > We thank the reviewer for pointing out the illuminating work [3].
> >
> > We note that the purposes of our paper and [3] are different.
> > As [discussed](https://openreview.net/forum?id=zz9hXVhf40&noteId=tiOvpIHe739) by the authors, [3] runs Bayesian Optimization (BO) on the online test environment, with the primary aim of providing insights into key design choices for offline MBRL, not to obtain state-of-the-art results or introduce a new methodology for realistic offline RL. By contrast, our paper provides an unified objective for model learning and policy improvement, and demonstrates some benefits of this approach over the baselines. Therefore, it may not be appropriate for us to use BO in the real environment to tune the hyperparameters of our methods, as in [3].
> >
> > Further, We notice that [3] uses the D4RL “v0” datasets [4] while we uses the latest version of the D4RL datasets, i.e., “v2" version for the Gym-Mojoco datasets and “v1" version for the Maze2D and Adroit (Line 202-205). Therefore, the optimized hyperparameters in [3] may not be directly applicable to our baseline comparison. Further, as discussed in Appendix G (page 35) of [3], the BO routine is time-consuming, taking up ~200 hours over an offline dataset for the MOPO method. This computation budget is currently out of our scope and out of the timing of the rebuttal period.
> >
> > Nevertheless, the analysis work of [3] is absolutely important for the community. It will be interesting to adopt an analysis similar to [3] to investigate the key design choices of our unified model-policy learning framework and provide a thorough guidance to tune the key hyperparameters. We will definitely work on this in our future work.
> >
> > \
> > [3] Lu, Cong, et al. "Revisiting design choices in offline model based reinforcement learning." International Conference on Learning Representations. 2021.
> > \
> > [4] Justin Fu, Aviral Kumar, Ofir Nachum, George Tucker, and Sergey Levine. D4rl: Datasets for
> > deep data-driven reinforcement learning. arXiv preprint arXiv:2004.07219, 2020.

---

> > > ### Author Response · Authors · 2022-08-07
> > > **Following up with Reviewer dzbb**
> > >
> > > Dear Reviewer dzbb,
> > >
> > > Thank you for your constructive review!
> > >
> > > We hope that our response can address your concerns.
> > > We would like to kindly confirm if you still have concerns on the disconnection of our implementation from the derivation, and on the practical choices in our methods.
> > >
> > > If so, we would like to take the discussion period to respond to any of your remaining/new concerns and address them in our revision.
> > > If our response and revised manuscript have answered your questions, would you mind re-evaluating our work based on the updated information?

---

> > > ### Comment · Reviewer_dzbb · 2022-08-08
> > > **Re: Response to Reviewer dzbb**
> > >
> > > I'd like to thank the authors for the detailed response and I believe that the response has addressed most of my concerns. Therefore, I’m increasing my evaluation of the paper and recommend acceptance. Additionally, I also wish to point out some misunderstandings of my review.
> > >
> > > > We note that the purposes of our paper and [3] are different.
> > >
> > > I pointed out the paper mainly as a reference to show that empirical implementation matters a lot in model based offline RL methods, not asking the authors to apply bayesian optimization to tune the algorithm (disclaimer: I’m not affiliated with any of the authors of that paper.). That paper as well as many prior works have shown that practical implementations of model based offline RL algorithms matter a lot, so pure theoretical analysis is often insufficient to explain the empirical performance of algorithms. Therefore I do hope that the authors could discuss the particle implementation side more.

---

> > > > ### Author Response · Authors · 2022-08-08
> > > > **Thank you for raising the rating**
> > > >
> > > > Dear Reviewer dzbb,
> > > >
> > > > We feel glad that our responses are helpful and deeply appreciate your recommendation of accepting our paper.
> > > >
> > > > For our previous response to [3], as we discussed in the answer to **Q3**, we absolutely agree with the reviewer that empirical choices matter a lot in model based offline RL methods.
> > > >
> > > > In Appendix H of the revision, we provide additional ablation studies on several practical choices in our main method, such as changing the JSD regularizer to other forms of regularizer, changing the implicit policy to the Gaussian policy, removing rollout data in policy learning, and so on. Indeed, we observe that the practical difficulty in the sample-based estimate of the KL-dual makes the alternative KL-dual-based regularizers ineffective. Some other practical choices are also discussed in Appendix H of the revision.
> > > >
> > > > As a future work, we will certainly abide by [3] to investigate further other implementation choices in our proposed framework.
> > > >
> > > >
> > > >
> > > > \
> > > > [3] Lu, Cong, et al. "Revisiting design choices in offline model based reinforcement learning." International Conference on Learning Representations. 2021.

---

### Official Review · Reviewer_3XmH · 2022-07-11

**Rating:** 5
**Confidence:** 4
**Soundness:** 2 fair
**Presentation:** 3 good
**Contribution:** 3 good

**Summary:**

This work proposes a new algorithm for offline model-based reinforcement learning (MBRL). Based on a lower bound of the cumulative reward objective (Eq.(5)), the proposed method alternatively optimizes the policy and the transition model using a fixed dataset. To bypass the difficulties in the objective, the part of the lower bound is further approximated by the Jensen-Shannon divergence (JSD) so that GAN training can be applied. In addition, the objective includes a marginal importance weight (MIW) that is also being estimated during training. Experiments on the D4RL benchmark show that the proposed method can achieve better performance than existing methods in the literature.

**Questions:**

Although it is nice to have a unified objective, there are several questions/concerns regarding the method.

1. The stability of the proposed method becomes a question when it alternates between learning several models (policy, action-value function, MIW, transition kernel, and the GAN discriminator). Each of them requires some sort of stability tricks, so the overall procedure seems very fragile as a whole.

2. L121: Why implicit policy is preferred? Gaussian or Beta policies are common in the literature.

3. When optimizing (13), 1. When optimizing (13), does some of the experience in the augmented dataset have fake rewards? Algorithm 1 does not mention anything about the learning of \hat{r} and how it is used.

4. The learning of MIW is interesting. Eq.(9) in the current paper is Eq.(4) in Wen et al., (2020) [44] when expanded. Such connection should be discussed and referenced. The main difference is that the current paper uses a test function. However, the motivation for using the action-value function as the test function is unsubstantiated and requires further discussion/evaluation. It seems more natural to use the reward function instead when it is motivated by the primal-dual relation (Eq.(1)), because we care more about learning accurate MIW. In this way, the LHS of (10) will be exactly the cumulative reward objective.

5. Experiments
- It would be more readable if the best model-based method is italicized/underlined instead of also bold in Table 1
- How are the MIWs normalized in Fig.2? It is surprising that the median of DualDICE can be so far away from 1 after normalization.
- The description in Sec.4.2(c) is unclear. The first sentence suggests that the regularization uses KL as in Thm.1, but the second sentence immediately suggests using JSD instead, which seems to be the “Main” variant. Specifically, how is WPR different from Main?

Minor
- L63: \pi_b is presumably the behaviour policy?
- L163: does not undermine the convergence property (under mild conditions on Q).

**Limitations:**

The authors have sufficiently addressed the limitations.

**Strengths And Weaknesses:**

Strengths
- A unified objective for training both the model and the policy
- Solid experiments

Weaknesses
- Lacking theoretical justification
- Some of the approximation steps are questionable

---

> ### Author Response · Authors · 2022-08-02
> **Response to Reviewer 3XmH (Part 1)**
>
> We thank the reviewer for your valuable comments and suggestions. Following are our responses to your detailed questions.
>
>  > **Q1:** The stability of the proposed method is questionable when alternating between learning several models. Each of them requires stability tricks, so the overall procedure seems very fragile.
>
>
>
> **A:** Our framework indeed requires more components than the naive model-based methods. However, the additional components can improve the overall performance. We further conducted ablation study, where we remove each component (e.g., regularization, model rollout, MIW) in our framework to check whether each component is necessary. The result is in Appendix H of the revision. The empirical results confirm that each component in our framework can improve the empirical performance.
> Besides,  we disagree that the current procedure is fragile. We have made efforts to make each component stable, either adapting previous stabilization techniques (GAN and action-value function) or new algorithms to stabilize training. As a result, we actually have a stable algorithm that achieves much better performance than those methods that do not consider the new components.
>
>
>  > **Q2:** Why is implicit policy preferred instead of the common Gaussian or Beta policies?
>
> **A:**
> As discussed in Section 3.2, the class of implicit policy is richer than the class of Gaussian/Beta policy, and thus can approximate more complex distributions. This can be helpful in multi-modal datasets such as Adroit or maze2d. Further, a number of prior works have demonstrated the advantages of using implicit policies in behavior cloning [1], offline model-free RL [2], and offline model-based RL [3].
>
> We further conducted an additional experiment by replacing the implicit policy in our Main method to the Gaussian policy. The results are presented in Table 5 and discussed in Appendix H of the revision. From Table 5, it is clear that using the Gaussian policy is less performant than using the implicit policy, especially on the Maze2D and Adroit datasets. This aligns with the finding in [2] that a uni-model Gaussian policy may not be rich enough to capture the necessary multiple action-modes in the Maze2D and Adroit datasets, while an implicit policy is likely capable.
>
> \
> [1] Florence, Pete, et al. "Implicit behavioral cloning." Conference on Robot Learning. PMLR, 2022.
> \
> [2] Yang, Shentao, et al. "A Regularized Implicit Policy for Offline Reinforcement Learning." arXiv preprint arXiv:2202.09673 (2022).
> \
> [3] Yang, Shentao, et al. "Regularizing a Model-based Policy Stationary Distribution to Stabilize Offline Reinforcement Learning." ICML, 2022.
>
>
>
>  > **Q3:** When optimizing (13), does some of the experience in the augmented dataset have fake rewards? How to learn $\hat{r}$? And how $\hat{r}$ is used?
>
>
> **A:**
> Thank you for pointing out this confusion!
>
> Yes. The synthetic data generated by the current policy $\pi_\phi$ and the learned dynamics all use the fake/learned rewards, as we do not assume that the reward functions are known. For simplicity, in the derivation of our method, we treat the reward $r$ as part of the (next) state of the transition model, as in MOPO-style methods [4-6].
>
> This concatenation of reward and (next) state also aligns with our practical implementation. As mentioned above and as discussed in Line 177-180 of Section 3.4, the reward function is optimized similarly as the transition dynamic, where the MIWs are used to reweight the MLE training loss.
>
> We clarify in Algorithm 1 of the revision the training and usage of the reward model $\hat r$.
>
> \
> [4] Yu, Tianhe, et al. "Mopo: Model-based offline policy optimization." Advances in Neural Information Processing Systems 33 (2020): 14129-14142.
> \
> [5] Yu, Tianhe, et al. "Combo: Conservative offline model-based policy optimization." Advances in neural information processing systems 34 (2021): 28954-28967.
> \
> [6] Lu, Cong, et al. "Revisiting design choices in offline model based reinforcement learning." International Conference on Learning Representations. 2021.
>
>  > **Q4:** Connection with Wen et al., 2020 should be discussed and referenced.
>
> Thanks for the suggestion. We have added a paragraph discussing VPM in Section 3.3.
>
>  > **Q5:** The motivation of using the action-value function as the test function is unsubstantiated and requires further discussion.
>
> We add an additional discussion in Appendix I  of the revision on why we choose to use the action-value function as the test function. Intuitively the action-value function is the optimal Lagrangian multiplier when we try to optimize the density ratio function in the Lagrangian function. We add a more rigorous mathematical explanation of this choice in Appendix I.
>
>  > **Q6:** Readability for Table 1.
>
> **A:**
> Thanks for the suggestions. We have modified Table 1 to underline the best of model-based methods if different from the overall best.

---

> > ### Author Response · Authors · 2022-08-02
> > **Response to Reviewer 3XmH (Part 2)**
> >
> > > **Q7:** How are the MIWs normalized in Fig.2? It is surprising that the median of DualDICE can be so far away from 1 after normalization.
> >
> > **A:**
> > As discussed in Section 4.2 (b), for numerical stability, the estimated MIW from DualDICE is clipped into $(10^{-8}, 500)$. This is implemented by adding differentiable softplus activations onto the output layer of the MIW network.
> >
> > As discussed in the caption of Figure 2, we plot the normalized MIWs whose mean on the entire dataset is 1. This is implemented by the code `w = w / w.mean(axis=0)`, where `w` is a tensor for the MIWs of the observations in the offline dataset.
> >
> > Figure 3 (Appendix G) in the revision shows the distribution plots of the $\log(\text{MIW})$ of the entire dataset generated by our method, and by the variants with the three alternative MIW estimation methods in Section 4.2 (b) over the training process on the example at Figure 2. In particular, in Figure 3, we see that for the DualDICE variant, the distribution of MIW on the entire dataset gradually degenerates on very small and very large values. By contrast, the MIWs from our method are well-shaped and concentrate around the mean 1 over the entire training process. We refer to Appendix G of the revision for a more detailed discussion on these distribution plots. To conclude, these plots show that our method can still perform well even when the current policy is far away from the behavior policy.
> >
> >
> > > **Q8:** Unclear description of Section. 4.2(c) and What is the difference between WPR and Main?
> >
> > **A:**
> > We have modified Section 4.2 (c) to make it more clear in the revision.
> >
> > In WPR, we keep the MIW for the policy-regularization term, which is the same as in $D_{\pi}(P^{*}, \widehat{P})$, thus we refer to it as weighted policy regularizer (WPR). However, we find it does not improve the performance, because when we estimate the regularization term in WPR, we incorporate weights into the minimax optimization of the policy and the discriminator, which may bring additional instability. As a result, we removed the MIW for policy regularization, which significantly improved the stability as we demonstrated in the empirical experiments.
> >
> >
> >
> > > **Q9:** Inaccurate writings and typos.
> >
> > **A:**
> > Thanks for pointing out the typos. We fixed it in the new version.

---

> > > ### Author Response · Authors · 2022-08-07
> > > **Following up with Reviewer 3XmH**
> > >
> > > Dear Reviewer 3XmH,
> > >
> > > We deeply appreciate your thoughtful review and your time, and hope that our response can address your concerns.
> > >
> > > May we kindly ask if you still have concerns on the stability of our method, our choice of the test function, or have any further concerns?
> > >
> > > We will be more than happy to address your remaining or new concerns and possibly revise our manuscript during the discussion period.
> > > If our response and revised manuscript have addressed your concerns, would you mind considering re-evaluating our work based on the updated information?

---

> > > ### Comment · Reviewer_3XmH · 2022-08-08
> > > **Additional comments**
> > >
> > > I want to thank the authors for the response and clarifications. I still have some concerns about stability and test function.
> > > - I didn't mean that different components are not necessary for the method. Instead, I meant that it might be hard to tune so many hyperparameters for new environments in practice.
> > > - As for the test function, the new Appendix I is not persuasive enough. $\omega$ appears in both terms of the Lagrangian and it remains unclear why we focus on the second term. One can also argue that the first term should be the focus and use the reward function as the test function. Additional discussion would be helpful.

---

> > > > ### Author Response · Authors · 2022-08-08
> > > > **Response to additional comments from Reviewer 3XmH**
> > > >
> > > > Dear Reviewer 3XmH,
> > > >
> > > > We deeply appreciate your raising of the concerns, which would definitely help us to improve the paper significantly. Belows are our response to your specific concerns.
> > > >
> > > > > **Concern1:** It might be hard to tune so many hyperparameters for new environments in practice.
> > > >
> > > > **A:** As discussed in Appendix D.2.1 (page 24), many of the hyperparameters in our method directly come from the literature **without further tuning**.
> > > >
> > > > Due to the diverse nature of offline datasets, prior works in offline model-based RL typically perform gentle dataset-specific hyperparameter tuning.
> > > >
> > > > As discussed in Appendix G.2 of MOPO [1], MOPO tunes the rollout length $h$ and the penalty coefficient $\lambda$ for each MuJoCo datasets.
> > > >
> > > > COMBO [2], another SOTA offline model-based RL baseline, tunes more hyperparameters. As discussed in Appendix B.2 of COMBO [2],  COMBO tunes for each MuJoCo datasets: rollout length $h$, Q-function and policy learning rates, conservative coefficient $\beta$, choice of $\rho(s,a)$, choice of $\mu(a \mid s)$, and the real data percentage $f$.
> > > >
> > > > By contrast, as shown in Appendix  D.2.1, we mainly tune three hyperparameters: rollout horizon $h$, noise dimension for the generator, and the smoothing exponent $\alpha$ in the MIW network. To minimize tuning, we have fixed $\alpha=0.5$ for all MuJoCo datasets and $\alpha=0.2$ for all the non-MuJoCo datasets. Other hyperparameters, including the penalty coefficient, all the learning rates, the real data percentage $f$, are fixed across all tested datasets.
> > > >
> > > > In particular, on the classical MuJoCo domain, we follow the same protocol as MOPO to tune two hyperparameters. Further, the number of hyperparameters that we tune is much less than COMBO.
> > > >
> > > > To sum up, compared with common offline model-based RL baselines, we believe we do not introduce more hyperparameters that possibly require tuning.
> > > >
> > > >
> > > > \
> > > > [1] Yu, Tianhe, et al. "Mopo: Model-based offline policy optimization." Advances in Neural Information Processing Systems 33 (2020): 14129-14142.
> > > > \
> > > > [2] Yu, Tianhe, et al. "Combo: Conservative offline model-based policy optimization." Advances in neural information processing systems 34 (2021): 28954-28967.
> > > >
> > > >
> > > >
> > > >
> > > >
> > > > > **Concern2:** Additional discussion on the test function.
> > > >
> > > > **A:** We update a new version of our paper, where we add the motivation and more discussions of choices of test functions in Appendix I. Specifically,  we add the following discussions (marked as red in the new version):
> > > >
> > > > - Motivations of why we want to get rid of minimax (saddle-point) based optimization approaches such as DualDICE and GenDICE.
> > > > - Discussion on VPM based approach and how it chooses the test function, also the advantages of VPM over DualDICE and GenDICE.
> > > > - A more detailed discussion on why we choose the action-value function as the test function, and a discussion of why we focus on the second term rather than the whole term (first and the second)
> > > > - Another intuitive explanation on why we choose action-value function as the test function (based on the speciality of off-policy evaluation problem).
> > > > - We include a remark to clarify the reason why we want to choose test functions to be meaningful and with nice properties.  In addition, we clarify that we may choose any functions to be the test function (including reward function as Reviewer 3XmH suggested), though these choices do not have some theoretical properties such that they may not help us to optimize $\omega$ easier in practice.
> > > >
> > > > Finally, as a demonstration, we just launched a new experiment by setting the reward function as the test function, without any other code modifications. The empirical results we will include in our next version if we can finish the experiments before the discussion period ends.
> > > >
> > > > -----
> > > > Updates: We added the empirical results as a demonstration in the followed up comment: https://openreview.net/forum?id=5yjM1sQ1uKZ&noteId=alj6nbgFbeS.

---

> > > > > ### Author Response · Authors · 2022-08-09
> > > > > **Results on using the reward as the test function**
> > > > >
> > > > > Dear Reviewer 3XmH,
> > > > >
> > > > > As a follow up to our previous response,
> > > > > Table A below compares the performance of our main method that uses the Q-function as the test function when training the MIW $\omega$, and a variant that uses the reward function as the test function.
> > > > >
> > > > > We agree with the reviewer that using the reward function as the test function is a possible alternative. However, from Table A, it is clear that this alternative generally has worse performance and larger standard deviations compared to the Main method. As we discussed in Appendix I of the latest manuscript, using the reward function as the test function loses the nice mathematical properties, and the resulting loss for the training of density ratio may not be easy to optimize.
> > > > >
> > > > >
> > > > > Table A: Comparison of our Main method and using the reward function as the test function in training the MIW $\omega$. Results shown are the mean $\pm$ std over 5 seeds.
> > > > > | Task   Name               | Main  (Q-function)            | Reward test-func. |
> > > > > |---------------------------|------------------|-------------------|
> > > > > | maze2d-umaze              | 55.8 $\pm$ 3.9   | 34.9 $\pm$ 24.3   |
> > > > > | maze2d-medium             | 107.2 $\pm$ 45   | 85.6 $\pm$ 45     |
> > > > > | maze2d-large              | 190.8 $\pm$ 34.2 | 143.8 $\pm$ 44    |
> > > > > | halfcheetah-medium        | 51.7 $\pm$ 0.4   | 51.5 $\pm$ 0.6    |
> > > > > | walker2d-medium           | 83.1 $\pm$ 1.8   | 77.8 $\pm$ 8.4    |
> > > > > | hopper-medium             | 58.9 $\pm$ 7.9   | 57.3 $\pm$ 9.7    |
> > > > > | halfcheetah-medium-replay | 44.6 $\pm$ 0.7   | 44.6 $\pm$ 0.2    |
> > > > > | walker2d-medium-replay    | 81.5 $\pm$ 3     | 72.8 $\pm$ 10.4   |
> > > > > | hopper-medium-replay      | 91.1 $\pm$ 9.5   | 79.3 $\pm$ 20.6   |
> > > > > | halfcheetah-medium-expert | 90.2 $\pm$ 2     | 88.5 $\pm$ 4.5    |
> > > > > | walker2d-medium-expert    | 107.7 $\pm$ 2.1  | 103 $\pm$ 6.9     |
> > > > > | hopper-medium-expert      | 93 $\pm$ 19.7    | 59.8 $\pm$ 26     |
> > > > > | pen-human                 | 20.6 $\pm$ 10.7  | 12.2 $\pm$ 10.6   |
> > > > > | pen-cloned                | 57.4 $\pm$ 19.2  | 37.5 $\pm$ 16.6   |
> > > > > | pen-expert                | 138.3 $\pm$ 6.6  | 133 $\pm$ 12.7    |
> > > > > | door-expert               | 96.3 $\pm$ 9.8   | 86 $\pm$ 24.3     |
> > > > > | Average Score             | 85.5             | 73                |
> > > > >
> > > > >
> > > > > ***
> > > > >
> > > > > We wish our further discussion in Appendix I and the above new empirical results can address your remaining concerns.  Could you kindly re-evaluate our work based on these new discussions?

---

> > > > > ### Comment · Reviewer_3XmH · 2022-08-09
> > > > > **Response**
> > > > >
> > > > > Thank you for the additional explanation and results. The new discussions in Appendix I and experiments with the reward function are interesting and convincing. As a result, I increased my score.
> > > > >
> > > > > Since the second term is supposed to be close to zero, I would further suggest showing its (empirical) value along the training to justify the stability of the algorithm.

---

> > > > > > ### Author Response · Authors · 2022-08-09
> > > > > > **Response Follow up for Reviewer 3XmH**
> > > > > >
> > > > > > Thank you very much for the response!  Since the discussion period is close to the end, below is a quick answer to your question:
> > > > > >
> > > > > > The second term is quite similar to the “Bellman residual error” for value function learning. If we use the target network to stabilize the training, then the loss calculated with target network can not reflect the value of the second term directly because the optimization loss (with target network) is not the true error for density ratio learning. This is exactly the same case as we train value functions using target networks (or fitted based algorithms), and evaluating the bellman residual error itself is a quite challenging problem in the value function learning literature. Similarly, to accurately evaluate the **true** value of the second term (the error) is also challenging.
> > > > > >
> > > > > > But we agree with Reviewer 3XmH that it would be useful to justify the stability of the algorithm. We think Figure 3 in the paper already provides some evidence that our approach is more stable than the other methods (DualDICE and GenDICE), and the empirical results also indicate our discussion on the stability of these methods in Appendix I. Besides, checking the overall distribution of density ratio values is similar to checking the value distribution of  value functions as used in the TD3 paper.
> > > > > >
> > > > > > As a result, we think we have already provided **evidence** that our method to train the MIW is quite stable.
> > > > > >
> > > > > > ------
> > > > > > Further, if Reviewer 3XmH thinks we have already addressed all your concerns, we would appreciate it if you could provide a more optimistic assessment of our work.

---

### Author Response · Authors · 2022-08-02
**General Response**

We want to thank all reviewers for their valuable suggestions. We submit a new version and highlight the changed part in blue with the following revisions:
1. We have added Appendix G to show the training dynamics of MIW produced by several methods (DualDICE, GenDICE, VPM and ours). And we briefly discuss the experimental results.
2. We have added additional ablation studies in Appendix H, including several variants of our proposed framework:
    - Removing some necessary components (MIW, regularizer, model-rollouts) respectively.
    - Changing JSD to KL to demonstrate that optimizing KL would introduce instability.
    - Changing implicit policies to Gaussian policy. (suggested by Reviewer 3XmH ).
3. We have added a discussion on why we choose action-value function as test function in Appendix I.
4. We have added a discussion on VPM (Wen et al. 2020) and our approach in Section 3.3.
5. We have fixed minor typos, wording and citation issues suggested by reviewers, and bolded results Table 1.

---

### Comment · Area_Chair_GW55 · 2022-08-06
**Please read the authors' response and start the discussion**

Dear Reviewers,

Thanks for providing the review. The discussion stage will end in next Tuesday. Please check the authors' response and feel free to discuss with authors.

Best, AC

---

### Meta-Review · Area_Chair_GW55 · 2022-08-30

**Recommendation:** Accept
**Confidence:** Less certain

**Metareview:**


In this paper, the authors motivated from the observation that the learning objectives of model and the polices in offline model-based RL are mismatched, and established a lower bound of the true expected return, which includes both the model and policy learning. The authors designed a tractable approximation to the lower bound, building upon which an algorithm is derived with practical implementation. The algorithm is then justified empirical to demonstrate the superior.

Most of the reviewers appreciate the proposed method. The paper can be further improved:

>All reviewers believes there are some gaps in the derivation of the practical algorithm from the original optimization. The rationale behind the approximation and parametrization should be clearly discussed to avoid the possible confusion. I understand there is page limit, but I believe the paper can be reorganized to leave the important discussion, especially the comparison w.r.t. DICE family and VPM, in the maintext, not in the Appendix. \


In sum, the paper indeed provided an interesting method which interactively learn the policy and the model in offline setting by optimizing the lower bound of the true expected return, and a practical approximation scheme with superior performance. I recommend for acceptance.

**Award:**

No

---

### Decision · Program_Chairs · 2022-09-14

Accept